# Flagellar energetics from high-resolution imaging of beating patterns in tethered mouse sperm

Ashwin Nandagiri[1,2,3], Avinash Satish Gaikwad[4], David L Potter[5], Reza Nosrati[3], Julio Soria[3], Moira K O'Bryan[6], Sameer Jadhav[2], Ranganathan Prabhakar[3]*

[1]IITB-Monash Research Academy, Mumbai, India; [2]Department of Chemical Engineering, Indian Institute of Technology Bombay, Mumbai, India; [3]Department of Mechanical and Aerospace Engineering, Monash University, Clayton, Australia; [4]School of BioSciences, University of Melbourne, Parkville, Australia; [5]Monash Micro-Imaging, Monash University, Clayton, Australia; [6]School of BioSciences, University of Melbourne, Parkville, Australia

**Abstract** We demonstrate a technique for investigating the energetics of flagella or cilia. We record the planar beating of tethered mouse sperm at high resolution. Beating waveforms are reconstructed using proper orthogonal decomposition of the centerline tangent-angle profiles. Energy conservation is employed to obtain the mechanical power exerted by the dynein motors from the observed kinematics. A large proportion of the mechanical power exerted by the dynein motors is dissipated internally by the motors themselves. There could also be significant dissipation within the passive structures of the flagellum. The total internal dissipation is considerably greater than the hydrodynamic dissipation in the aqueous medium outside. The net power input from the dynein motors in sperm from *Crisp2*-knockout mice is significantly smaller than in wildtype samples, indicating that ion-channel regulation by cysteine-rich secretory proteins controls energy flows powering the axoneme.

*For correspondence:
prabhakar.ranganathan@monash.edu

**Competing interests:** The authors declare that no competing interests exist.

## Introduction

In their journey towards the oocyte, sperm propel themselves by beating a whip-like flagellum. This motility is essential for successful fertilization and is fundamental to reproduction. Understanding sperm motility is essential for improving male infertility treatments, animal breeding, and wildlife conservation (*Gaffney et al., 2011*). Despite the vast body of work on the structure and function of different parts of the axoneme – the internal 'engine' powering the flagellum (*Brokaw and Kamiya, 1987*; *Okagaki and Kamiya, 1986*; *Yagi et al., 2005*) – and other accessory structures that surround the axoneme, such as the outer dense fibers (*Zhao et al., 2018*) and the fibrous sheath (*Eddy et al., 2003*), the mechanisms that control the complex beating patterns observed in flagella remain poorly understood (*Brokaw, 2009*; *Lehti and Sironen, 2017*; *Lindemann and Lesich, 2016*; *Lin and Nicastro, 2018*). It is, however, recognized that mechanical properties of the flagellum and its surroundings play a crucial role in determining sperm motility (*Gaffney et al., 2011*). Measurements of the mechanical behavior of single flagella in living sperm have however remained a critical bottleneck.

We demonstrate here a set of powerful new tools that enable detailed calculation of the mechanical energetics of single sperm flagella from high-resolution optical microscopy. Automated image-analysis tools have long been used to study sperm movement (*Katz et al., 1975*; *Katz and Overstreet, 1981*; *Overstreet et al., 1979*). Computer-aided sperm analysis systems are today used extensively in clinical settings to rapidly assess the viability of samples containing hundreds of cells in a single field of view (FOV) (*Amann and Waberski, 2014*). These high-throughput techniques,

however, do not resolve flagellar motion. Improvements in digital imaging and storage have now placed within reach the high-speed, high-resolution, and long-exposure imaging that researchers of flagellar propulsion have long sought (*Gray, 1955*; *Gray, 1958*; *Brokaw, 1966*; *Rikmenspoel et al., 1960*). A wide range of digital image processing algorithms are now available (*Gonzalez et al., 2004*) that can be combined with high-performance parallel computing to analyze thousands of video frames with little manual intervention (*Baba and Mogami, 1985*; *Riedel-Kruse et al., 2007*; *Saggiorato et al., 2017*; *Hansen et al., 2018*; *Sartori et al., 2016*). We have implemented these image-analysis techniques to automatically extract centerlines of sperm flagella in every video frame.

To quantitatively analyze beat patterns in a statistically meaningful way, we need to image swimming sperm over several beat cycles. While rapid progress is being made on full three-dimensional tracking (*Muschol et al., 2018*; *Dardikman-Yoffe et al., 2020*; *Gadêlha et al., 2019*), it is unlikely that sufficient beat cycles can be reliably recorded with freely swimming sperm that can quickly move out of focal plane or the FOV (*Mondal et al., 2020*). Instead, we image flagella beating freely in the focal plane in cells tethered chemically at their heads to a glass slide. Our tethered-cell assay, in principle, permits imaging single cells until they stop beating. We report here results obtained by analyzing large numbers of (~50) beat cycles in single tethered sperm in freshly prepared samples when they are most vigorous (*Gaikwad et al., 2020*).

Beating patterns in sperm flagella have been studied previously to investigate changes induced by environmental factors (*Bukatin et al., 2015*; *Smith et al., 2009*; *Saggiorato et al., 2017*) or by gene mutations (*Krähling et al., 2013*; *Lim et al., 2019*). We build here on the suggestion that the technique of proper orthogonal decomposition (POD) can be applied on the time-resolved tangent-angle profiles of flagellar centerlines to analyze their kinematics (*Ma et al., 2014*; *Werner et al., 2014*; *Saggiorato et al., 2017*). POD is widely applied in the analysis of turbulent flows (*Lumley, 1967*; *Holmes et al., 2012*) and other fields (*Baumberg and Hogg, 1994*; *Jolliffe, 2002*) to reduce complexity of spatiotemporal patterns and represent them with a much smaller set of numbers, while still retaining accuracy. To objectively compare flagellar beating patterns, we apply POD to unambiguously identify the *mean beat cycle* of each sperm from the time series of the tangent-angle profiles of its flagellar centerline. We can compute average cycles of any kinematic or dynamic quantity derived from the tangent-angle profiles. We further introduce a technique to consistently represent the POD shape modes with smooth Chebyshev polynomials to ensure that the tangent-angle profile is sufficiently smooth and its spatial derivatives can be computed without spurious artifacts. The tangent-angle profile obtained thus is consistent with the rigid-body kinematics of the stiff head region. This Chebyshev-POD (C-POD) technique allows for efficient calculation of geometric quantities such as the local curvature and kinematic quantities such as the velocity components, at any material point on the centerline.

Our approach for calculating forces and energetics from the measured beating patterns stems from ideas discussed originally by *Machin, 1963*. We use the geometric and kinematic data to determine the hydrodynamic resistance offered by the external fluid medium using resistive force theory (RFT) (*Gray and Hancock, 1955*; *Lighthill, 1976*) and further calculate internal forces by applying conservation principles. This requires a model for the mechanical behavior for the flagellar body. Several models have been proposed that consider the flagellum to be an 'active' material (*Camalet et al., 1999*; *Camalet and Jülicher, 2000*; *Lindemann, 1994a*; *Lindemann, 1994b*; *Sartori et al., 2016*; *Chakrabarti and Saintillan, 2019*). These are based on different models for motor forcing in the axoneme and the regulation of their kinetics. We propose instead a different approach that is agnostic to the nature of motor activity and avoids invoking the assumption that the flagellar material is active. We consider the motion of the non-motor passive material of the flagellum under the action of the unknown forces exerted by the axonemal motors. This allows us to use well-established principles for the continuum material stress in the passive flagellar material. The resulting Soft-Internally -Driven-Kirchhoff-Rod (SIDKR) model leads to an energy balance across the flagellum, which we then use to determine the spatiotemporal distribution of motor power across the flagellum over its mean cycle.

We have used this approach to analyze flagellar beating patterns of sperm from wildtype (WT) and *Crisp2* knockout (KO) mice. The cysteine-rich secretory proteins (CRISPs) are a group of proteins that are predominantly expressed in the male reproductive tract (Gaikwad et al., in preparation). *Crisp2* is incorporated into the sperm acrosome, connecting piece and the outer dense fibers of the sperm tail. It is known that the deletion of *Crisp2* in mice leads to compromised sperm function,

including altered sperm motility (*Hu et al., 2018*; *Lim et al., 2019*). The precise effect on flagellar function, however, is unknown.

Our observations with these sperm reveal intriguing new information: there is considerable intra-cellular friction within the flagellum. This challenges the widely held view that the hydrodynamic resistance offered by the viscous fluid medium outside is the sole dissipative sink that must be over-come by the continual driving provided by the dynein motors. Further, the flagellar filament is also conventionally regarded as an elastic body that perfectly stores energy temporarily by bending. Our findings suggest instead that internal friction within the passive structures of the flagellum, and within the motors themselves, may be as large as the external hydrodynamic friction. These are in line with recent observations also made in algal cilia (*Mondal et al., 2020*). These sources of internal dissipation could therefore play a significant role in determining beating patterns in sperm (*Camalet and Jülicher, 2000*). This insight could be vital for understanding dramatic changes in fla-gellar beating patterns induced by changes in the medium (*Smith et al., 2009*) or the proximity of surfaces (*Nosrati et al., 2015*; *Denissenko et al., 2012*).

## Theoretical model
### The soft, internally driven Kirchhoff rod model

Flagellar motion is driven internally by the action of dynein motors distributed within the axoneme. The sperm body is treated as a slender, flexible filament immersed in a viscous fluid (*Figure 1*). It is assumed that the passive material of the sperm body is a Kirchhoff rod (*Audoly and Pomeau, 2010*; *Malvern, 1969*; *O'Reilly, 2017*), that is, it is inextensible and each of its material cross-sections remains rigid and planar, while rotating with respect to each other about the rod axis as it bends and twists. The passive Kirchhoff rod has external as well as internal surfaces. It is driven by axone-mal motors acting on its internal surfaces and the resulting motion is resisted by the hydrodynamic forces that act on its external surface (*Figure 1*) as well as the stresses that arise to resist material deformation as the rod bends.

The instantaneous space curve of the axial centerline of the filament, $\mathbf{r}(s,t)$, is parameterized by its arc length variable, $s$, defined such that $s = 0$ at the tip of the head, and $s = L$ at the tail end. A local material frame is attached to each cross-sectional plane and is specified by a triad of unit vec-tors, $\mathbf{d}_k$, where $k = 1, 2, 3$. In general, the smooth variation of these vectors with $s$ at any instant of time, $t$, is specified in terms of the Darboux vector, $\Omega$, where $\partial \mathbf{d}_k / \partial s = \Omega \times \mathbf{d}_k$. The components $\Omega_k$ of the Darboux vector are the generalized curvatures. Since we shall only consider motion of the rod in the $x - y$ plane, we align the material frame at each cross-section with the Frenet–Serret frame

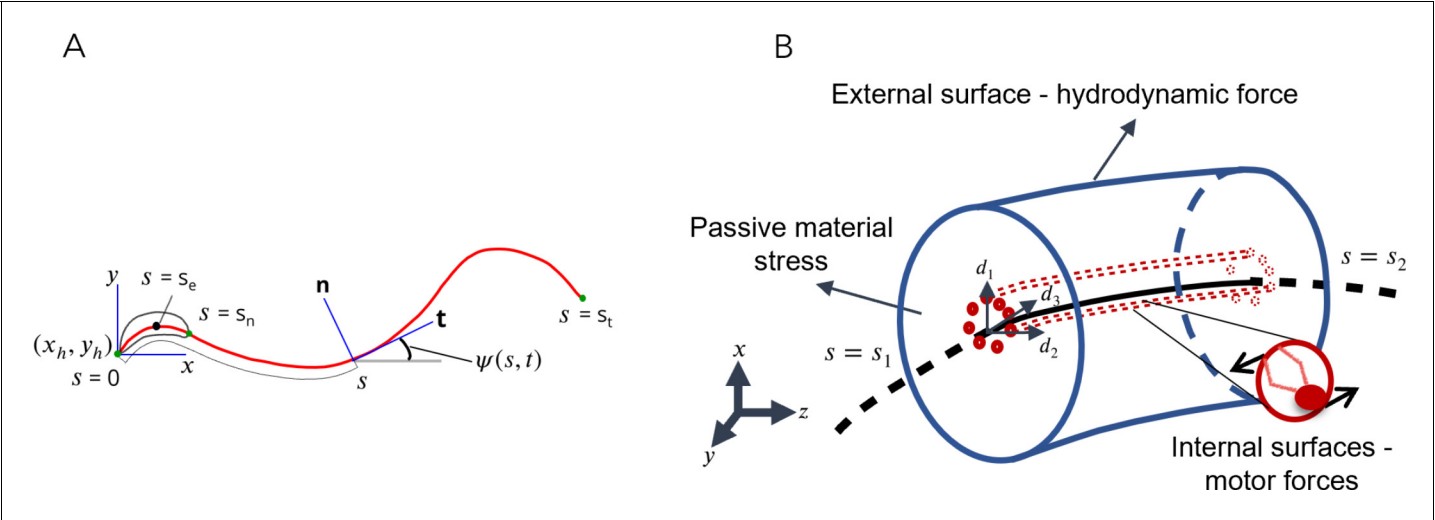

**Figure 1.** Schematic representations for the Soft, Internally driven Kirchhoff rod model. (**A**) Geometric variables defined along the centerline. (**B**) An arbitrary control volume used for deriving the equations of the model: the volume consists of the passive flagellar material; hydrodynamic forces act on the external surface while axonemal motors act on the internal surfaces. The passive material adjacent to the cross-sectional faces at either end exerts stresses on those faces.

associated with each point on the axial curve. For this choice, $\mathbf{d}_1 = \mathbf{t} = \partial \mathbf{r}/\partial s$, the unit tangent vector to axial curve. The other two vectors, $\mathbf{d}_2 = \mathbf{n}$ and $\mathbf{d}_3 = \mathbf{b}$, are the normal and binormal vectors, which span the cross-sectional plane. The Darboux vector for the Frenet–Serret frame is $\Omega = T(s,t)\,\mathbf{d}_1 + C(s,t)\,\mathbf{d}_3$, where $C$ and $T$ are the curvature and torsion profiles at any time. For planar motion, $\mathbf{b} = \mathbf{e}_z$ (pointing out of the plane of the page) is a constant; hence, $T = 0$. The geometry of a planar Kirchhoff rod at any instant is thus fully specified by the curvature, $C$. The velocity of a point on the centerline, $\mathbf{v}(s,t) = \partial \mathbf{r}/\partial t$. Cross-sectional planes can rotate relative to each other. Then, $\partial \mathbf{d}_k/\partial t = \omega \times \mathbf{d}_k$, where $\omega(s,t)$ is the instantaneous angular velocity of a cross-sectional plane at $s$. It can further be shown that $\omega$ and $\Omega$ satisfy the compatibility relation (**Powers, 2010**),

$$\frac{\partial \omega}{\partial s} = \sum_{i=1}^{3} \frac{\partial \Omega_i}{\partial t}\,\mathbf{d}_i \,. \tag{1}$$

For planar motion, where $\omega = \omega \mathbf{e}_z$,

$$\frac{\partial \omega}{\partial s} = \frac{\partial C}{\partial t} \,. \tag{2}$$

For inertialess rods, consideration of the conservation of linear momentum for a segment of the rod where $s \in [s_1, s_2]$ formally yields the following equation (see Appendix 1):

$$\mathbf{f}^{\mathrm{a}} + \mathbf{f}^{\mathrm{h}} + \mathbf{f}^{\mathrm{e}} + \frac{\partial \mathbf{F}}{\partial s} = \mathbf{0} \,, \tag{3}$$

where $\mathbf{f}^{\mathrm{a}}(s,t)$ and $\mathbf{f}^{\mathrm{h}}(s,t)$ are the force distributions per unit length on the cross-section at any $s$ due to the surface tractions exerted by internal motor activity and the external hydrodynamic resistance, respectively. Other external forces, such as the force exerted by a tethering traction at a wall, are accounted for by the distribution $\mathbf{f}^{\mathrm{e}}(s,t)$. The passive stress in the Kirchhoff rod results in a force, $\mathbf{F}$, exerted on a cross-section by the material on its aft side. The gradient with respect to $s$ of $\mathbf{F}$ in the momentum balance thus describes the *net* restoring force per unit length on a cross-section due to passive internal stresses resisting deformation. From conservation of angular momentum, we obtain (Appendix 1):

$$\mathbf{m}^{\mathrm{a}} + \mathbf{m}^{\mathrm{h}} + \mathbf{m}^{\mathrm{e}} + \mathbf{t} \times \mathbf{F} + \frac{\partial \mathbf{M}}{\partial s} = \mathbf{0} \,, \tag{4}$$

where $\mathbf{m}^{\mathrm{a}}(s,t)$ and $\mathbf{m}^{\mathrm{h}}(s,t)$ are the torques per unit length exerted by the surface tractions due to the internal motors and the external viscous hydrodynamic resistance; $\mathbf{m}^{\mathrm{e}}$ is the torque distribution due to other external forces. The torque on a cross-section exerted by the passive material stresses on its aft side is $\mathbf{M}$, and its gradient in the equation above is the net restoring torque distribution. Energy conservation further shows that at any cross-section, in general,

$$\frac{\partial \epsilon}{\partial t} + \frac{\partial u}{\partial t} = p^{\mathrm{a}} + p^{\mathrm{hd}} + p^{\mathrm{e}} + p^{\mathrm{s}} - q \,, \tag{5}$$

where $\epsilon(s,t)$ is the local elastic energy per unit length (i.e., the elastic storage density) of the rod and $u(s,t)$ is the thermal internal energy density. On the right-hand side, $q$ is the net rate of heat removal per unit length of the rod by the surroundings, while each of the remaining terms is, respectively, the mechanical power per unit length delivered into the rod cross-section by the action of the motors, the hydrodynamic and non-hydrodynamic external forces, and the passive material stress. The motor power distribution, $p^{\mathrm{a}}$, is the key unknown in our study. The hydrodynamic power distribution is related to the corresponding force and torques distributions:

$$p^{\mathrm{hd}} = \mathbf{v} \cdot \mathbf{f}^{\mathrm{h}} + \omega \cdot \mathbf{m}^{\mathrm{h}} \,. \tag{6}$$

The other external mechanical power $p^{\mathrm{e}}$ is similarly related to the external force and moment distributions, $\mathbf{f}^{\mathrm{e}}$ and $\mathbf{m}^{\mathrm{e}}$. The net rate of work done on a cross-section by the action of the local stress gradient is

$$p^{\mathrm{s}} = \frac{\partial\,(\mathbf{v}\cdot\mathbf{F})}{\partial s} + \frac{\partial\,(\omega\cdot\mathbf{M})}{\partial s}\,. \tag{7}$$

The sign convention used here is that mechanical power due to work done *on* a cross-section of the rod and tending to increase the local internal energy storage is positive whereas the power due to work done by that cross-section to overcome resistances leading to a decrease in stored energy is negative. Due to its purely dissipative nature, $p^{\mathrm{hd}}$ is therefore always negative at any $s$ and $t$. In our study, the external force and moment due to the tethering constraint exerted on the head cannot be measured directly. The mechanics of this tether could be complex and, at any instant of time, $p^{\mathrm{e}}$ may be positive or negative. However, over a full cycle, we expect net work to be done by the cell against the tethering constraint. The key advantage in treating the motor contribution as a forcing that is external to the passive material of the Kirchhoff rod is that we can treat the active forcing as an unknown to be extracted from experimental data in a model agnostic manner while applying well-established concepts to treat passive material stresses within the Kirchhoff rod. The passive stress tensor can be formally split into an elastic part and a part that provides internal dissipation, so that the total material torque, $\mathbf{M} = \mathbf{M}^{\mathrm{el}} + \mathbf{M}^{\mathrm{id}}$. It can be shown that *Equation (62)* is satisfied when the elastic torque arising from the passive material stress is such that

$$\frac{\partial\epsilon}{\partial t} = \mathbf{M}^{\mathrm{el}}\cdot\frac{\partial\omega}{\partial s} = \mathbf{M}^{\mathrm{el}}\cdot\frac{\partial\Omega}{\partial t}\,, \tag{8}$$

and the dissipative part of the material stress is such that

$$\frac{\partial u}{\partial t} = \mathbf{M}^{\mathrm{id}}\cdot\frac{\partial\omega}{\partial s} - q = -p^{\mathrm{id}} - q\,, \tag{9}$$

where $p^{\mathrm{id}}$ denotes the rate of internal frictional dissipation per unit length. Since the material of the Kirchhoff rod is passive, the Second Law of Thermodynamics requires that $p^{\mathrm{id}} \leq 0$ everywhere (*Chaikin and Lubensky, 1995*). Since the dynein motors are excluded from the control volume in the analysis above, $p^{\mathrm{id}}$ does not include any dissipation that occurs within the motors themselves. We shall later discuss how we separately obtain the motor dissipation.

## Constitutive relations

Although presented in the context of a sperm body, the equations above are generally valid of any inertialess, internally driven Kirchhoff rod. To proceed further, we make several constitutive assumptions that are specific to the case of a sperm cell tethered at its head. The sperm body is assumed to be composed of a head region, $s \in [0, s_{\mathrm{N}}]$, and a flagellar tail region, $s \in (s_{\mathrm{N}}, L)$, with $s_{\mathrm{N}}$ denoting the location of the neck junction between the two regions. We assume that the head is a rigid body. In our experiments, cells are further tethered at a point in the head region, and the head can rotate rigidly about this tether point. Therefore, although the angular velocity $\omega \neq \mathbf{0}$ in the head region, rigid-body kinematics dictates that $\partial\omega/\partial s = \mathbf{0}$ everywhere in the head region. Hence, from *Equation (48)*, $\partial\Omega_k/\partial t = 0$ across the head. Therefore, for planar beating, $\partial\omega/\partial s = \partial C/\partial t = 0$ across the head. The flagellar tail is flexible and not subject to the kinematic constraints above.

The head does not contain internal motors, which are all distributed only along the tail region. Therefore, $\mathrm{f}^{\mathrm{a}}$, $\mathrm{m}^{\mathrm{a}}$, and $p^{\mathrm{a}}$ are all zero for $s \in [0, s_{\mathrm{N}}]$. In the flagellar tail, each dynein motor is assumed to act on the internal surfaces of a cross-section such that the forces exerted at its two ends are of equal magnitude but in opposite directions. Therefore, $\mathbf{f}^{\mathrm{a}} = \mathbf{0}$. However, the net torque they exert is not zero, and therefore $\mathbf{m}^{\mathrm{a}} \neq \mathbf{0}$, which serves to drive the filament's motion. The external hydrodynamic force distribution is given by RFT (*Gray and Hancock, 1955*; *Lighthill, 1976*):

$$\mathrm{f}^{\mathrm{h}} = -[\zeta_t\,\mathrm{tt} + \zeta_n\,(\delta - \mathrm{tt})]\cdot\mathrm{v}\,, \tag{10}$$

where the tangential and normal hydrodynamic friction coefficients in an infinite fluid medium of viscosity, $\mu$, are $\zeta_t = 2\pi\mu/\ln(2L/a)$ and $\zeta_n = 4\pi\mu/[\ln(2L/a) + 1/2]$, respectively. For sperm tethered to a glass slide, the no-slip condition at the slide surface creates an additional resistance to fluid flow. *Katz et al., 1975* obtained the following RFT approximations for the friction coefficients for motion of a slender body in a plane parallel to a wall and at a distance of $h$ from it:

$$\zeta_t = \frac{2\pi\mu}{\ln 2\,h/a}; \quad \zeta_n = \frac{4\pi\mu}{\ln 2\,h/a}. \tag{11}$$

These coefficients have previously been used in a number of studies, notably by Jülicher and co-workers (*Riedel-Kruse et al., 2007*) for analyzing experimental data on wall-tethered sperm and, more recently, by *Mondal et al., 2020*, for tethered axonemes isolated from cilia. The cross-sectional radius of the cylindrical filament, $a$, is further not constant along the sperm body. For our calculations here, only the variation of the radius in the tail region is relevant. We assume a linear taper along the flagellum, that is, for $s \geq s_{\mathrm{N}}$,

$$a(s) = (a_{\mathrm{N}} - a_{\mathrm{T}})\frac{L-s}{L-s_{\mathrm{N}}} + a_{\mathrm{T}}. \tag{12}$$

where $a_{\mathrm{N}}$ and $a_{\mathrm{T}}$ are the radii at the neck and the tail tip. When a sperm tethered at its head beats in a plane parallel to the wall, $h = a_{\mathrm{N}}$, is constant (Appendix 2).

The rigid head region requires no further constitutive assumptions. The tail region can deform and therefore requires a constitutive model that relates its material stresses to its deformation. The simplest constitutive model for the elastic stress in a passive material is the Hookean model, which leads to a linear relation between the elastic material torque and the local curvature. The corresponding elastic energy distribution must be consistent with *Equation (8)*. Thus, in the tail region,

$$M_i^{\mathrm{el}} = \sum_{i=1}^{3} \kappa_i \Omega_i \mathrm{d}_i; \quad \epsilon = \sum_{i=1}^{3} \frac{\kappa_i \Omega_i^2}{2}. \tag{13}$$

where $\kappa_i$ is an elastic stiffness coefficient. The simplest constitutive model for the dissipative stress that satisfies the condition imposed by the Second Law that the dissipation rate is always positive leads to the following expression for the dissipative part of the internal torque:

$$\mathbf{M}^{id} = \eta\,\frac{\partial\omega}{\partial s}, \tag{14}$$

where $\eta > 0$ is the internal friction coefficient per unit length. Taken together, the constitutive equations above are equivalent to modeling the Kirchhoff rod as a passive viscoelastic Kelvin–Voigt solid (*Bird et al., 1987*). For the linear taper assumed in the tail region, the elastic stiffness and internal friction coefficients can be shown to vary with the radius as $a^4$. That is,

$$\kappa(s) = \kappa_{\mathrm{N}}\left(\frac{a(s)}{a_{\mathrm{N}}}\right)^4 \quad \eta(s) = \eta_{\mathrm{N}}\left(\frac{a(s)}{a_{\mathrm{N}}}\right)^4 \tag{15}$$

where $\kappa_{\mathrm{N}}$ and $\eta_{\mathrm{N}}$ are the values of the elastic stiffness and frictional coefficients at the neck. For planar motion, $\mathbf{M} = M\mathbf{e}_z$, and the relations above reduce to

$$M^{\mathrm{el}}(s,t) = \kappa(s)\,C; \quad \epsilon(s,t) = \kappa(s)\frac{C^2}{2}; \quad M^{\mathrm{id}}(s,t) = \eta(s)\frac{\partial\omega}{\partial s}. \tag{16}$$

We make a few other simplifying assumptions. The head and tail ends are free; $\mathbf{F}$ and $\mathbf{M}$ are, therefore, zero at the two ends. The external surface traction due to tethering at the wall acts at a single location, $s_{\mathrm{E}}$, on the head and is zero elsewhere, that is, $\mathbf{f}^{\mathrm{e}} = \mathbf{F}^{\mathrm{e}}\delta(s - s_{\mathrm{E}})$ and $\mathbf{m}^{\mathrm{e}} = \mathbf{M}^{\mathrm{e}}\delta(s - s_{\mathrm{E}})$. The system is further isothermal and changes in the internal thermal energy of the body are negligible, that is, $\partial u/\partial t = 0$ in *Equation (62)* and *Equation (9)*. This means that any internal frictional heat generation is, therefore, instantaneously balanced by, $q$, the heat removal from the passive flagellar material to its surroundings. Further, the ratio of the contributions from the external hydrodynamic moment, $\mathbf{m}^{\mathrm{h}}$, and the hydrodynamic force, $\mathbf{f}^{\mathrm{h}}$, to the total hydrodynamic power, that is, the ratio $|\omega \cdot \mathbf{m}^{\mathrm{h}}|/|\mathbf{v} \cdot \mathbf{f}^{\mathrm{h}}|$, is expected to scale as $a/L \ll 1$. The contribution of $\mathbf{m}^{\mathrm{h}}$ in *Equation (60)* to the hydrodynamic dissipation is, therefore, neglected. The momentum and energy balance equations for the rigid, passive, head region on which the external tether force acts, and the viscoelastic, untethered, internally driven tail region are summarized in Appendix 1. We next describe our approach to quantifying the kinematics of the beating patterns recorded in experiments and then using these along with the momentum and energy balances to obtain the dynamics and energetics of sperm.

## Kinematics from image analysis and POD

In Materials and methods, we describe in detail the image-analysis and data-processing algorithms used to obtain power distributions from microscope videos of tethered sperm samples from WT and *Crisp2* KO mice. Briefly, the image-analysis algorithm is used to process videos of single sperm cells tethered to a glass surface and beating in the focal plane of the microscope and extract centerlines of sperm bodies in every video frame. This raw data is first analyzed for head region separately to determine its motion as a rigid body. Twentieth-order Chebyshev polynomials are fitted through these centerlines to construct smooth tangent-angle profiles (see *Figure 1A*) of the flagellar tail region. These Chebyshev polynomials are designed to be consistent with the rigid-body kinematics of the head region.

In general, the POD is an order-reduction technique that optimally approximates spatiotemporally varying data. In our C-POD approach, we apply POD on the time-dependent Chebyshev coefficients to represent the deviation of $\psi(s,t)$, the time-resolved tangent-angle profile of the centerline from its time average, $\psi_0(s)$, as a weighted sum of $M$ orthogonal shape modes (see C-POD of the tail region). In other words,

$$\psi(s,t) = \psi_0(s) + \sum_{m=1}^{M} B_m(t)\psi_m(s). \tag{17}$$

The set of 'shape modes', $\psi_m$, $m = 1 \ldots M$, is optimal in the sense that, for any given $M$, the approximation above is guaranteed to deviate least from the original data than any other expansion in terms of another set of $M$ mutually orthogonal basis functions (*Holmes et al., 2012*; *Werner et al., 2014*). We describe, in Materials and methods, the C-POD method to obtain the shape modes, each of which is a 20th order, Chebyshev polynomial that is consistent with the head region executing rigid-body rotation. The corresponding time-dependent weights of the shape modes are referred to as 'shape coefficients'. With the smooth C-POD tangent profiles, we can efficiently compute at any $s$ and $t$, geometric and kinematic quantities in the beating plane, such as the curvature $C$ and its derivatives with respect to $s$ or $t$, the flagellar velocity $\mathbf{v}$, and the cross-sectional angular rotation rate, $\omega$.

## Dynamics and energetics from measured kinematics

The hydrodynamic force distribution, $\mathbf{f}^{\mathrm{h}}$, is first calculated using *Equation (10)* and the expressions for the tangential and normal friction coefficients. Using *Equation 3* together with the boundary condition that $\mathbf{F}(L,t) = 0$ at the tail tip, we then obtain

$$\mathrm{F}(s,t) = \int_{s}^{L} \mathrm{f}^{\mathrm{h}}(s',t)\,ds', \tag{18}$$

for all $s$ in the tail region. The moments, $M^{\mathrm{el}}$ and $M^{\mathrm{id}}$, and the elastic energy density $\epsilon$ are calculated using the constitutive *Equations (13) and (14)* and the elastic stiffness and internal dissipation profiles, $\kappa(s)$ and $\eta(s)$, in *Equation (15)* along with the values of the parameters, $\kappa_{\mathrm{n}}$ and $\eta_{\mathrm{n}}$. The total bending moment, $M = M^{\mathrm{el}} + M^{\mathrm{id}}$.

The energetic variables are then calculated as follows. In the tail region, the rate of change of the elastic storage density, $\epsilon$, and the power dissipated due to internal friction per unit length are (from *Equations 8 and 9*), respectively,

$$\frac{\partial \epsilon}{\partial t} = \kappa(s)\,C\,\frac{\partial C}{\partial t}; \quad p^{\mathrm{id}} = -\eta(s)\left(\frac{\partial \omega}{\partial s}\right)^2. \tag{19}$$

We henceforth denote the rate of elastic storage density as $\dot{\epsilon}$. The external hydrodynamic dissipation due to flagellar motion, $p^{\mathrm{hd}}$, is calculated using $\mathbf{v}$ and $\mathbf{f}^{\mathrm{h}}$ in *Equations (60)*. Consistent with their dissipative natures, $p^{\mathrm{hd}}$ and $p^{\mathrm{id}}$ are always negative. The gradient in the mechanical power due to the internal force and bending moment, $p^{\mathrm{s}}$, is obtained using *Equation (61)*. There are no other external forces acting on the freely beating tail. The external power distribution, $p^{\mathrm{e}}$, is therefore zero at all points in the tail region. The energy balance, *Equation (62)*, can be rearranged as follows for the tail region:

$$p^{\mathrm{a}}(s,t) = \dot{\epsilon} - p^{\mathrm{s}} - p^{\mathrm{hd}} - p^{\mathrm{id}} . \tag{20}$$

The active power distribution along the tail can be obtained with all the terms on the right-hand side determined from centerline kinematics as described above.

The integrals of each term in the equation over the entire tail region give the instantaneous net rates of change of the energetic variables. For instance, the net instantaneous storage rate, $\dot{E}(t) = \int_{s_{\mathrm{N}}}^{L} \dot{\epsilon} \, ds$. The instantaneous total hydrodynamic and passive internal frictional dissipation rates, $P^{\mathrm{hd}}$ and $P^{\mathrm{id}}$, and the net active power, $P^{\mathrm{a}}$, are similarly calculated by integrating the distributions $p^{\mathrm{hd}}$, $p^{\mathrm{id}}$, and $p^{\mathrm{a}}$ over the tail region, respectively. We can similarly obtain rates over just the mid-piece or over the principal piece alone. We further define and calculate

$$P^{\mathrm{md}}(t) = \int_{s_{\mathrm{N}}}^{L} \min(p^{\mathrm{a}},0) \, ds ; \quad P^{\mathrm{mi}}(t) = \int_{s_{\mathrm{N}}}^{L} \max(p^{\mathrm{a}},0) \, ds . \tag{21}$$

As we shall show later, the active power distribution is not always positive, and $P^{\mathrm{md}}$, the integral of $p^{\mathrm{a}}$ over its negative values is the total rate at which energy is dissipated within the dynein motors themselves. We will show below that, $P^{\mathrm{mi}}$, the integral over the positive values of $p^{\mathrm{a}}$ is the actual instantaneous power input from the dynein motors into the filament that is necessary to overcome all the different sources of dissipation. We shall refer to $P^{\mathrm{md}}$ and $P^{\mathrm{mi}}$ as the motor dissipation and the motor input, respectively.

Besides the various sources of energy dissipation in the tail region, there is also dissipation against the hydrodynamic and tethering forces acting across the head region. Since the head is modeled as a rigid, passive body, there is no elastic storage or internal dissipation in that region, nor is there any active motor power. Thus the work required to move the head against the hydrodynamic and tethering forces must come from the force, $\mathbf{F}_{\mathrm{N}}$, and the moment $M_{\mathrm{N}}$, exerted by the flagellum on the head at the neck junction. Hence, the instantaneous power dissipated by the head against the hydrodynamic and external tethering forces,

$$P_{\mathrm{H}}^{\mathrm{hd}} + P_{\mathrm{H}}^{\mathrm{e}} = P_{\mathrm{H}}^{\mathrm{d}} = -(\mathbf{v}_{\mathrm{N}} \cdot \mathbf{F}_{\mathrm{N}} + \omega_{\mathrm{N}} M_{\mathrm{N}}) , \tag{22}$$

the power delivered on to head by the force acting on the neck junction. Since $\mathbf{F}$ and $\mathbf{M}$ must be continuous across the neck junction, $\mathbf{F}_{\mathrm{N}} = \mathbf{F}(s_{\mathrm{N}},t)$ using *Equation (70)* and $M_{\mathrm{N}} = M^{\mathrm{el}}(s_{\mathrm{N}},t) + M^{\mathrm{id}}(s_{\mathrm{N}},t)$ calculated using *Equation (16)*.

The physical boundary conditions at the tail end of the flagellum are $\mathbf{F}(L,t) = 0$ and $M(L,t) = 0$. Integrating *Equation (71)* over the entire tail region with these boundary conditions at the tail end, and *Equation (66)* at the head end, and noting that $P^{\mathrm{a}} = P^{\mathrm{md}} + P^{\mathrm{mi}}$, we obtain, at any $t$,

$$P^{\mathrm{mi}} = \dot{E} - P_{\mathrm{H}}^{\mathrm{d}} - P^{\mathrm{hd}} - P^{\mathrm{id}} - P^{\mathrm{md}} . \tag{23}$$

The time averages of these instantaneous power functions over a *single cycle* are referred to as their 'cycle-means'. These cycle-means are denoted by an overline. Since the motion of the flagellum is periodic but noisy, there is no net storage of elastic storage in the flagellum over many cycles, that is, the average of the cycle-mean, $\overline{\dot{E}}$, over several cycles must be zero. The cycle-means of the dissipation rates are, however, not zero. Therefore, neglecting the fluctuations due to $\overline{\dot{E}}$, we calculate the cycle-mean of the motor input as the power input required to balance the dissipations due to head motion, external hydrodynamic resistance and internal friction, and the dissipation within the motors:

$$\overline{P}^{\mathrm{mi}} = -\left( \overline{P}_{\mathrm{H}}^{\mathrm{d}} + \overline{P}^{\mathrm{hd}} + \overline{P}^{\mathrm{id}} + \overline{P}^{\mathrm{md}} \right) . \tag{24}$$

The average of the cycle-means over all beat cycles is identically equal to the time average over the entire duration of observation and will be referred to as such and denoted by a double overline (e.g., $\overline{\overline{P}}^{\mathrm{hd}}$) .

In Results, we compare the relative magnitudes of these different dissipations. The results are obtained with the medium viscosity, $\mu = 10^{-3}$ Pa s. The radius at the neck and at the tail end are

$a_n = 0.57\,\mu m$ and $a_t = 0.18\,\mu m$, respectively (*Gu et al., 2019*). The total body length $L$ for each sample is taken to be the maximum observed length in the sample video and is around 120 μm. There are few measurements of the bending stiffness for sperm flagella in the literature. The stiffness of flagella in mouse sperm is reported to be between that of bull ($1.5 \times 10^3$ Pa μm$^4$) and rat ($3 \times 10^4$ Pa μm$^4$) sperm (*Lindemann and Lesich, 2016*). We use their geometric mean $7 \times 10^4$ Pa μm$^4$ as the

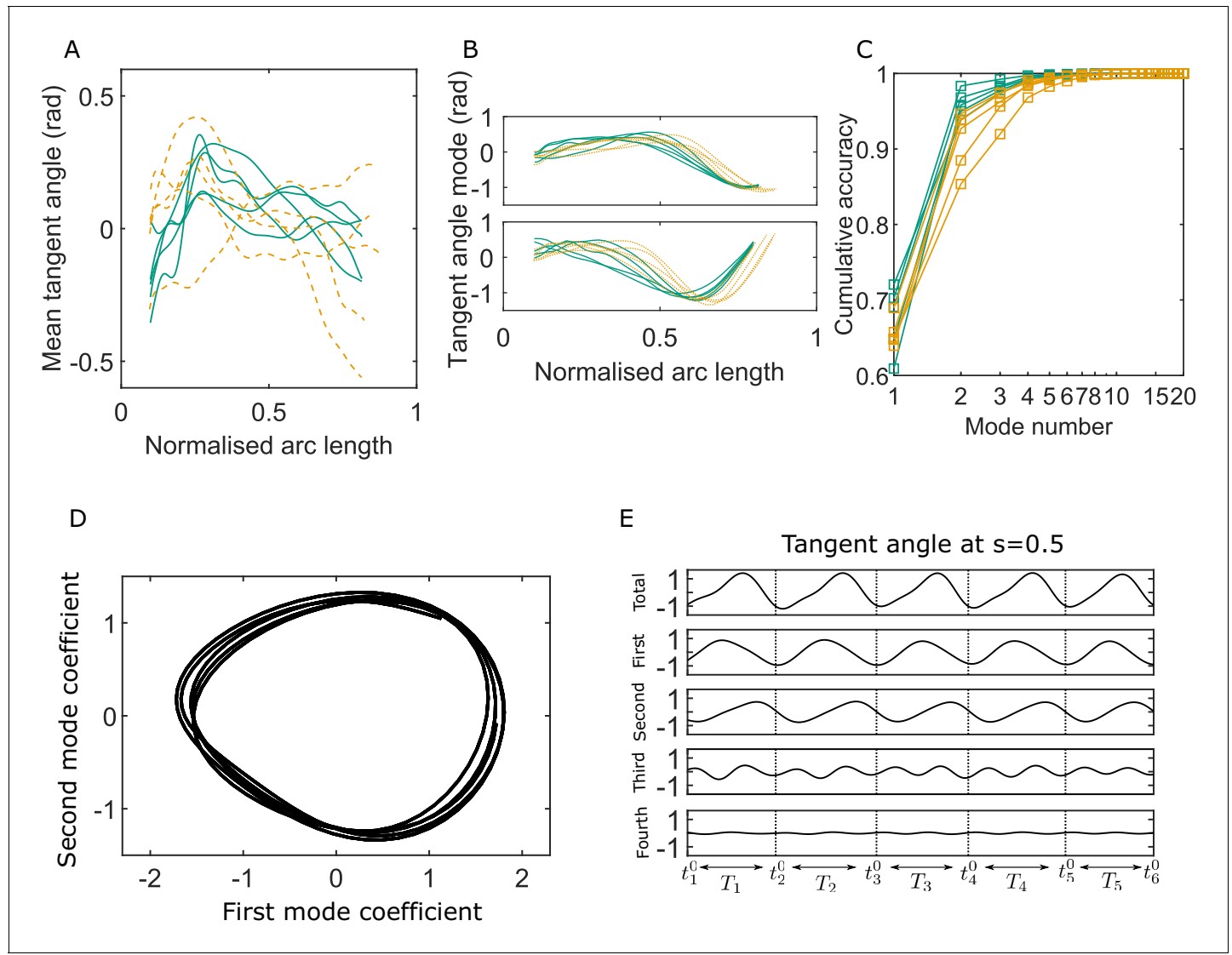

**Figure 2.** Key variables of the Chebyshev polynomial-based proper orthogonal decomposition (C-POD) of experimental tangent-angle data. (**A**) Time-averaged tangent-angle profiles for five wildtype (WT) (continuous curves) and five knockout (KO) (dashed curves) samples. (**B**) First (top) and second (bottom) C-POD shape modes for WT (continuous curves) and KO (dashed curves) samples; the colors are as in (**A**). (**C**) Cumulative accuracy of the C-POD representation for WT and KO samples; the colors are as in (**A**). A representation using the first four modes captures 95% or more of the observed centerline shapes for all samples. (**D**) Five shape cycles for a single WT sample in the parameter space defined by the time-dependent coefficients of the first two C-POD shape modes. The zero-crossing of the second modal coefficient marks the start of a new cycle. (**E**) Contributions of the first four modes to the tangent angle at the midpoint of the sperm body in the five tangent-angle cycles in (**D**): the horizontal line in the top plot is the time-averaged tangent angle for this WT sample. The starting time of the $i$ th cycle is denoted as $t_i^0$, and its duration (i.e., cycle time) is $T_i$.

The online version of this article includes the following source data for figure 2:

**Source data 1.** Numerical data for *Figure 2A, B*.
**Source data 2.** Numerical data for *Figure 2C*.
**Source data 3.** Numerical data for *Figure 2D*.
**Source data 4.** Numerical data for *Figure 2E*.

value for $\kappa_{\mathrm{N}}$ in calculations here. There are, however, no clear measurements yet of the internal bending friction coefficient, $\eta_{\mathrm{N}}$. We report below the results obtained for flagellar energetics with both $\eta_{\mathrm{N}} = 0$ and $10^3$ Pa s μm$^4$ and discuss the reasons why the latter value may be realistic.

## Results

### POD enables identification of beat cycles

*Figure 2* summarizes generic observations on the C-POD shape modes and their coefficients. In all the results presented here, the arc-length coordinate $s$ along the centerline is normalized by the maximum observable length of the whole flagellum in the entire duration of a sample video. The mid-piece region corresponds approximately to values of $s$ in the range 0.1–0.3, and the principal piece extends from $s = 0.3$ to $s = 0.85$.

Mouse sperm heads have distinctive falciform (hook) shapes (*Woolley, 2003*). In the image-processing protocol we have followed, all video frames are initially digitally rotated or reflected such that the head is on the left end of the body with the hook facing concave downward. For most of the WT and KO samples, the time-averaged tangent angles ($\psi_0(s)$) are observed in *Figure 2A* to consistently first increase with $s$ around the mid-piece region before decreasing in the principal piece. Since the local curvature $C = \partial\psi/\partial s$, the gradient of the tangent angle with respect to $s$, *Figure 2A* shows that the time-averaged shape for these samples is curved such that it is concave in the anti-hook direction in the mid-piece and concave in the pro-hook direction in the principal piece. The mean shapes thus show that the asymmetric spatial bias in the beating pattern over time is not uniform across the flagellum. In the one outlier KO sample (KO-5) in *Figure 2A*, however, the mean shape is anti-hook concave throughout. Intrinsic net asymmetry in flagellar beating is well known in sperm in many mammalian species, even when uncapacitated. Our observation that the mean shape is curved with an anti-hook (ventral) concave shape is consistent with the observations of *Woolley, 2003* that, in mouse sperm, the flagellum bends at the neck more on the ventral side than on the other.

The periodic beating of the flagellum about the mean shape is described by the C-POD shape modes and their time-dependent coefficients. The shapes of the first two shape modes ($\psi_1(s)$ and $\psi_2(s)$) in *Figure 2B* are qualitatively similar across the WT and KO samples. The key advantage of using the POD method to represent beating patterns is its optimality: a significant proportion of the beating pattern can be studied and understood by considering just a few shape modes. *Figure 2C* plots the cumulative contribution of the shape modes to the overall accuracy in capturing the full centerlines. Just the first two modes achieve a capture efficiency greater than 92% for all the WT samples, and for three out of the five KO samples. Even for the other two KO samples these dominant shape modes account for more than 85% of the observed beating patterns. Across all samples, the first four modes describe at least 95% of the beating patterns. We therefore calculate all kinematic, dynamic, and energetic quantities using the first four shape modes and their time-dependent coefficients.

As pointed out by *Werner et al., 2014* and *Ma et al., 2014*, the periodicity in the beating pattern is clearly brought out by plotting the coefficients $B_1(t)$ and $B_2(t)$ of the two dominant modes against one another. For any sperm sample, the trajectory traced out in $B_1$-$B_2$ phase space consists of loops, one for each beat cycle (e.g., *Figure 2D*). We choose here to demarcate the start and end time for each beat cycle as the time at which the polar angle in the $B_1$-$B_2$ phase space crosses zero. This choice means that, in each sperm sample, the shape at the start of a beat cycle always corresponds mostly to the shape of the first dominant mode (with minor contributions from modes higher than the second; *Figure 2E*). Thus, the overall time series for any quantity can be split into individual beat cycles, as demonstrated in *Figure 2E*. Although *Figure 2D, E* shows only a few cycles for clarity, the C-POD technique applied to tethered sperm makes it possible to systematically accumulate data for large numbers of beat cycles and quantitatively compare, in a statistically meaningful sense, individual sperm samples within a genotypical population and also compare one genotypical population with another.

## Active power distribution provides evidence for energy dissipation by dynein motors

We first present the spatiotemporal variations typically observed in all our samples in the energetic quantities. *Figure 3* plots the kymographs for the different energetic contributions obtained with the scaling estimate of the internal friction coefficient, $\eta_{\mathrm{n}} = 10^3$ Pa s $\mu$m$^4$, over several beat cycles for one of the WT samples. Similar results are obtained for all the other samples. The banded

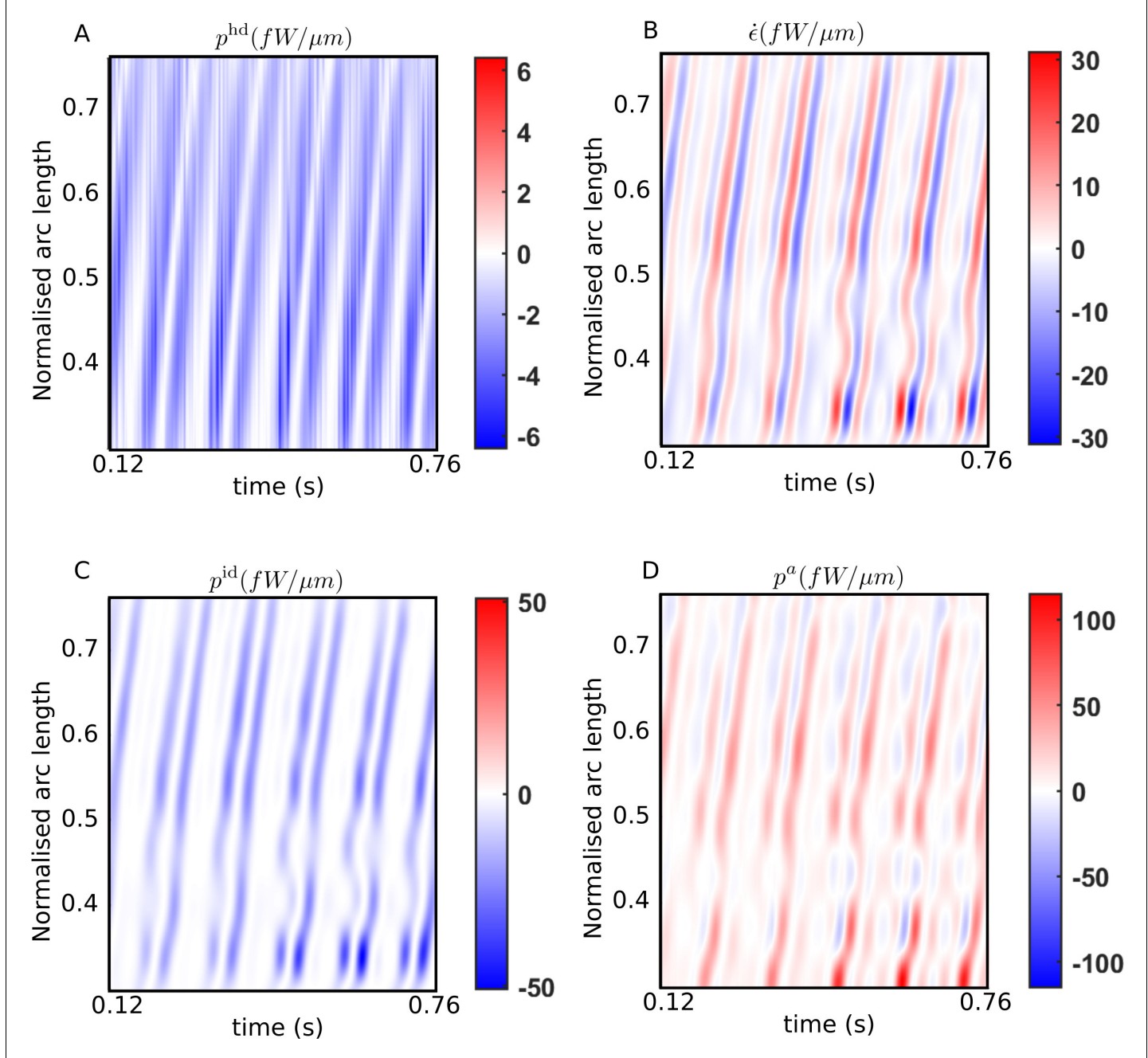

**Figure 3.** Spatiotemporal distributions of the rates of (A) hydrodynamic dissipation, (B) elastic storage, and (C) internal dissipation, and (D) the active power along the flagellum of a wildtype sperm over several beat cycles: red indicates positive rates, while blue indicates negative rates. The data in (C) and (D) have been obtained using the scaling value of $10^3$ Pa s $\mu$m$^4$ for the internal friction coefficient.

The online version of this article includes the following source data for figure 3:

**Source data 1.** Numerical data for all kymographs.

structures in these kymographs provide a visual confirmation of the spatiotemporal periodicity of the energy variables corresponding to the periodic beating of the flagellum.

In *Figure 3A*, the hydrodynamic power distribution, $p^{\mathrm{hd}}$, is always negative: that is, every part of the flagellum is at all times working against the hydrodynamic forces exerted externally by the viscous environment provided by the ambient fluid. This work done on the fluid is dissipated away by fluid friction. The elastic storage rate per unit length, $\dot{\epsilon}$ at any location $s$, however, alternates between positive (red) and negative (blue) values in *Figure 3B*. As a bending wave propagates through that location, the local curvature at that $s$ increases, leading to potential energy being stored elastically and a positive rate of $\dot{\epsilon}$ at that location. As the filament begins to relax and straighten out, the stored elastic energy is released and begins decreasing, leading to negative $\dot{\epsilon}$ values there. The filament then proceeds to bend in the other direction at that point, leading to a second positive growth of $\dot{\epsilon}$ within the same beat cycle, followed by a negative phase in $\dot{\epsilon}$ as the filament relaxes back towards being undeformed and straight at that location. Thus, at any $s$ in *Figure 3B*, each beat cycle consists of two successive positive and negative growth rate phases in $\dot{\epsilon}$.

Comparing the bands in *Figure 3B* with those in *Figure 3A*, it is clear that every single planar wave that propagates down the filament is associated with a pair of hydrodynamic dissipation peaks: the contribution of any single location to the hydrodynamic dissipation peaks as the filament moves quickly while bending and relaxing back on one side, and then again, on the other side. These bands are mirrored in *Figure 3C*, which plots $p^{\mathrm{id}}$, the distribution of power dissipated due to *internal* friction. This frictional dissipation, calculated with $\eta_{\mathrm{n}} = 10^3$ Pa s $\mu$m$^4$, is due to relative motion between adjacent cross-sectional planes of the flagellar material, which also peaks at a location when a bend towards one side or the other propagates past that point.

The external and internal dissipations and temporary elastic storage of energy must together be supported by the mechanical power input provided by the dynein motors acting on the microtubule surfaces of the flagellum. *Figure 3D* plots the distribution of the net active power density $p^{\mathrm{a}}$, across the filament. Interestingly, we find that the $p^{\mathrm{a}}$ distribution displays clear negative bands that repeatedly occur in all beating periods and are spread throughout the filament. The positive domains (red) of the $p^{\mathrm{a}}$ kymograph in *Figure 3D* represent mechanical power being delivered on the passive parts of the filament by the motors. In those regions, the motors cause relative sliding of microtubule doublets to rotate the local cross-sectional planes in the same sense as the torques they exert, that is, since $p^{\mathrm{a}} = \omega \cdot m^{\mathrm{a}}$, $p^{\mathrm{a}}$ is positive at a cross-section when both the rotational velocity of that plane, $\omega$, and the torque per unit length, $m^{\mathrm{a}}$, exerted by the dynein motors in that plane have the same sign. On the other hand, where $p^{\mathrm{a}}$ is negative (blue) in *Figure 3D*, $\omega$ and $m^{\mathrm{a}}$ are opposite in sign. At any such point, work is being done *by* the rest of the flagellar material *on* the axonemal motors, driving them back *against* the torque they continue to exert. We observe this behavior consistently in all beat cycles and for all WT and KO samples.

The energy transferred back as mechanical work on the motors can neither be stored either within the dyneins nor converted back to chemical free energy (i.e., ATP): it must be therefore quickly dissipated locally within the axoneme itself. This axonemal *motor dissipation* is measured by the negative domains of $p^{\mathrm{a}}$ and is denoted here as $p^{\mathrm{md}}$. This is a second source of dissipation within the flagellum and is distinct from the dissipation, $p^{\mathrm{id}}$, that is due to internal friction arising from the relative motions of all the other structures in the flagellum that surround the axonemal motors, such as the microtubules, the outer dense fibers, etc. By adding together the $p^{\mathrm{a}}$ distribution over all the locations where it is negative, we can calculate, $P^{\mathrm{md}}$, the instantaneous rate of energy dissipation due to the dynein motors themselves. The sum of $P^{\mathrm{md}}$ and $P^{\mathrm{id}}$ is the total mechanical power dissipated *within* the whole flagellum.

## POD enables statistics of beating patterns and energetic variables

The qualitative features of the distributions of the key energetic variables discussed above are common to both WT and KO samples. Before identifying significant differences between the beating patterns and energetics of the genotypes, it is worth examining the sample-to-sample variability within each population. *Figure 4A* shows the mean cycle of the beating pattern in physical *x-y* space for each sperm sample in our study. Flagellar centerlines at the same value of the fractional duration of the mean beat cycle have the same color in *Figure 4A*. This fractional duration of the mean cycle is referred to as the time phase and is denoted as $\tau$. To obtain the mean centerline shape at a

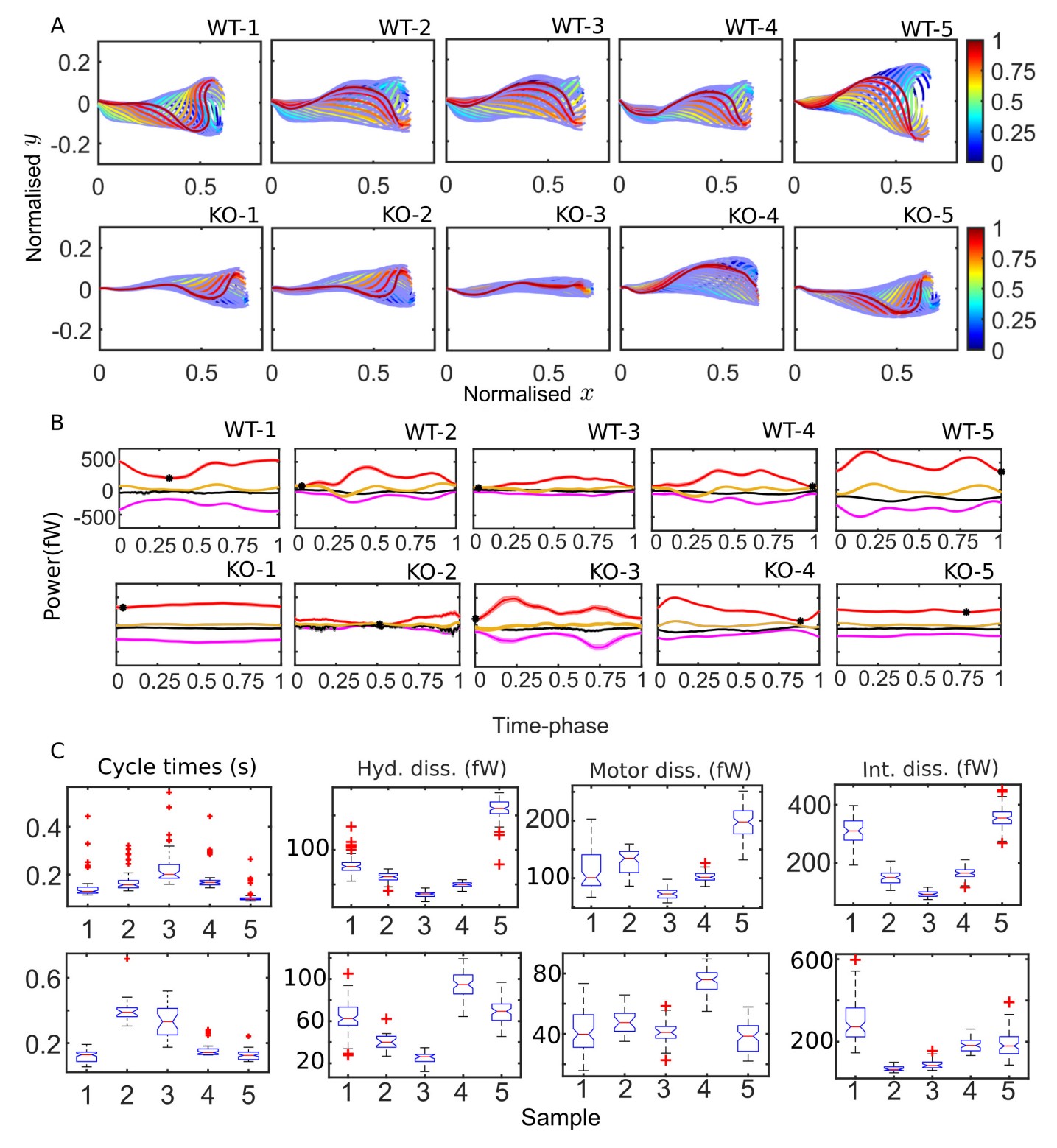

**Figure 4.** Mean cycles of beat patterns and energetics. (A) Each colored curve shows the mean shape at a particular phase of the mean cycle for the five wildtype (WT) (top row) and knockout (KO) (bottom row) samples. The color bands around each curve indicate the standard error in the mean component. (B) Mean cycles for the magnitudes of the net elastic storage (yellow), hydrodynamic dissipation (black), internal frictional dissipation (magenta), and active power (red) in WT (top panel) and KO (bottom panel) sperm samples corresponding to those in (A). Bands show standard errors in means. (C) Statistical distributions of cycle times and dissipation rates in each of the WT (top panel) and KO (bottom panel) samples. The box-plots

*Figure 4 continued on next page*

*Figure 4 continued*

present the median (red line), the first and third quartile (bottom and top box edges), and minimum and maximum (lower and upper whiskers) values for 40–60 cycles. Outliers that are more than 1.5 times the interquartile range away from the top or bottom of the box are indicated by red crosses. The notch extremes correspond to $q_2 \pm 1.57(q_3 - q_1)/\sqrt{n}$, where $q_1$, $q_2$, and $q_3$ are the first, second (median), and third quartiles, respectively, and $n$ is the number of observations (*McGill et al., 1978*).

The online version of this article includes the following source data for figure 4:

**Source data 1.** Numerical data for *Figure 4A*.
**Source data 2.** Numerical data for *Figure 4B*.
**Source data 3.** Numerical data for *Figure 4C* (cycle times).
**Source data 4.** Numerical data for *Figure 4C* (hyd. dissn., motor dissn., and int. dissn.).

particular value of $\tau$, we collect, at that $\tau$, the $x$ and $y$ coordinates obtained (using *Equation 44*) for all the beat cycles, and then calculate their mean values. The bands in *Figure 4A* around the mean centerlines are the standard errors in the mean (SEM) $y$ coordinates at each $s$. Our procedure for identifying the start and end of each beat cycle thus enables calculation and comparison of average beating patterns.

The difference between the mean beat patterns of the WT and *Crisp2* KO samples is striking. The KO samples exhibit a smaller amplitude across the entire flagellar tail. In *Figure 4B, C*, we apply the idea of calculating mean cycles to the energetic variables calculated from the four-mode C-POD of the tangent-angle profiles. *Figure 4B* compares the mean cycles in the *net* rates of elastic storage, ($\dot{E}$; yellow), hydrodynamic ($P^{\mathrm{hd}}$; black) and internal frictional ($P^{\mathrm{id}}$; magenta) dissipations, and the *net* rate of motor power input ($P^{\mathrm{a}}$; red curves). At each time phase, $\tau$, in a beat cycle, these mean rates are calculated by collecting the values of $\dot{E}$, $P^{\mathrm{hd}}$, $P^{\mathrm{id}}$, and $P^{\mathrm{a}}$ from all the cycles and averaging those values. No distinctive common patterns are immediately apparent across the WT or KO samples in *Figure 4B*. In 7 of the 10 samples, the minimum value in $P^{\mathrm{a}}$ ($P^{\mathrm{a}}_{\mathrm{min}}$; black symbols in *Figure 4B*) occurs close to the beginning or end of the cycle, when the first shape mode is dominant, suggesting that the first shape mode could be associated with a state of minimum power input. In two of the KO samples (KO-1 and KO-5), however, the net motor power remains nearly constant over the entire cycle.

It is visually apparent from *Figure 4B* that the mean cycles of the energy flows vary considerably from sample to sample. We plot the distributions of cycle times for each of the WT (*Figure 4C*; top panel, i) and KO (bottom panel, i) samples. Also shown as box-plots are the statistical distributions of the magnitudes of the cycle-averaged hydrodynamic, passive internal friction and motor dissipation powers. The cycle power in any single cycle is calculated by integrating an instantaneous power with respect to time over that cycle and dividing by the cycle time for that cycle. In the following sections, we use this data to answer two questions. Firstly, how large are the internal dissipations due to passive and motor friction relative to the external hydrodynamic dissipation? Secondly, what is the effect of the *Crisp2* gene deletion on flagellar energetics?

## Internal dissipation is larger than external hydrodynamic dissipation

The novel finding in *Figure 4B, C* is that, for any WT or KO sample, the magnitudes of the internal frictional and motor dissipations are comparable to or larger than the dissipation in the external fluid. Before we examine this further, it must be reiterated that the results in *Figure 4* for these dissipation rates depend on the values of the material parameters $\kappa_{\mathrm{n}}$ and $\eta_{\mathrm{n}}$. As previously mentioned, we have used here $\kappa_{\mathrm{n}} = 7 \times 10^4$ Pa $\mu$m$^4$ based on experimental measurements elsewhere (*Lindemann and Lesich, 2016*). While the existence of internal friction in the fluid-filled region around the axoneme is expected (*Riedel-Kruse et al., 2007*; *Mondal et al., 2020*), direct measurements of the value of $\eta_{\mathrm{n}}$ are not available.

For the same sperm motion quantified by the tangent-angle C-POD, we have calculated the energetics with different values of $\eta_{\mathrm{n}}$, ranging from zero to values well above the scaling estimate of $10^3$ Pa s $\mu$m$^4$. For any value of $\eta_{\mathrm{n}}$, we robustly find negative domains in the active power distribution, $p^{\mathrm{a}}$. However, as *Figure 5A* shows, for the 10 sperm samples studied, the minimum value of the net motor power delivered in a mean cycle, $P^{\mathrm{a}}_{\mathrm{min}}$, has a strongly negative value when $\eta_{\mathrm{n}}$ is much smaller than $10^3$ Pa s $\mu$m$^4$. For such values of $\eta_{\mathrm{n}}$, there is a significant portion of the mean cycle when $P^{\mathrm{a}}$ is

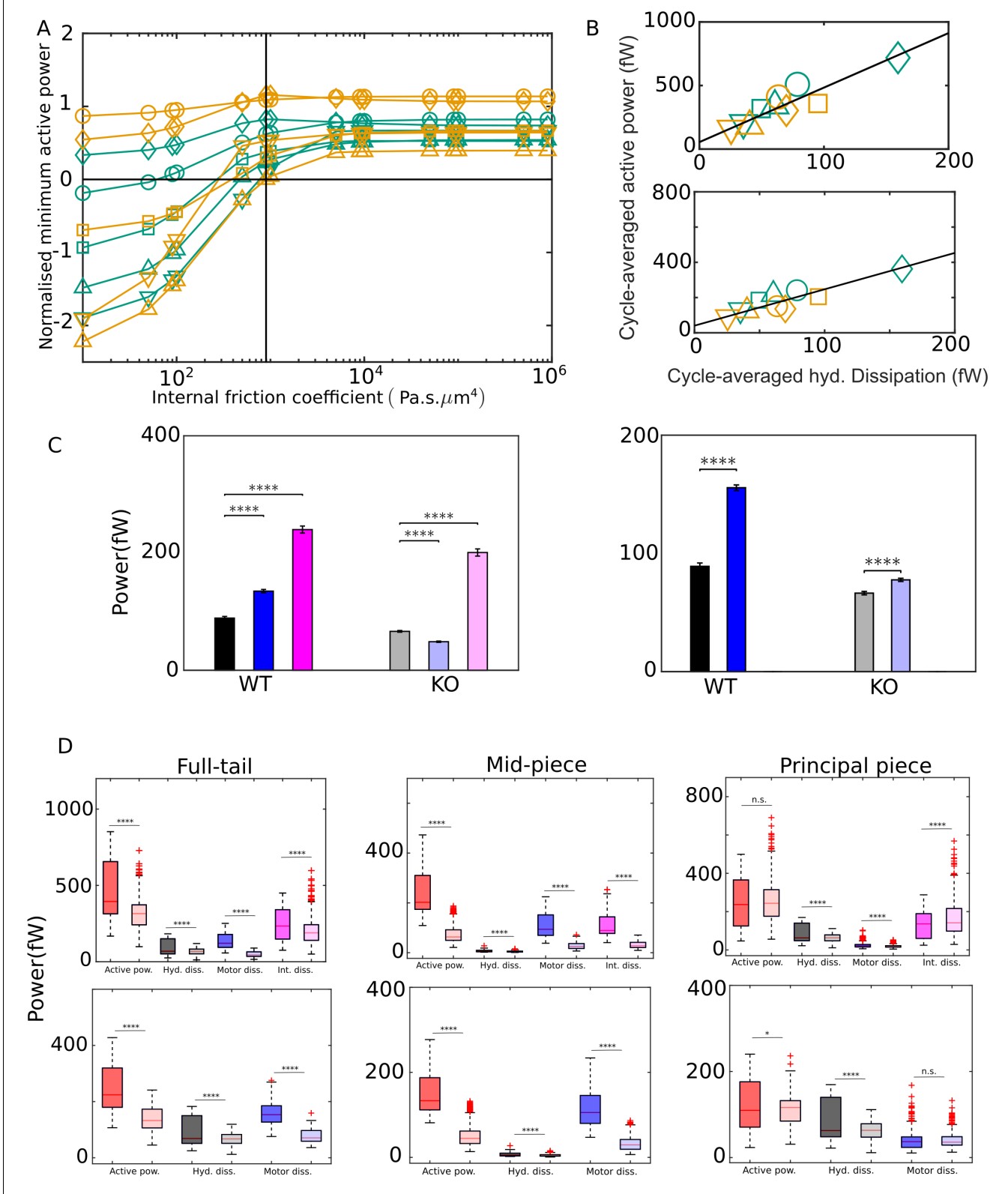

**Figure 5.** External versus internal dissipation in flagella. (**A**) Effect of the value of internal friction coefficient, $\eta_{\text{N}}$, on the minimum value of the net active power required to overcome dissipation for wildtype (WT) (blue) and knockout (KO) (red) samples. At each $\eta_{\text{N}}$, and for each sample, the minimum in the mean cycle of the net active power, $P_{\text{min}}$, is normalized by the time average of the motor input, $\overline{\overline{P}}^{\text{mi}}$, over all cycles. The vertical line is the scaling value

*Figure 5 continued on next page*

Figure 5 continued

of $10^3$ Pa s µm$^4$. (B) Correlation of time averages of the hydrodynamic dissipation and net active power for $\eta_N = 10^3$ Pa s µm$^4$ (top) and 0 (bottom). The lines are linear fits through data for both species. (C) Comparison of the external hydrodynamic dissipation (black) with the motor (blue) and passive internal frictional (magenta) dissipations obtained with $\eta_N = 10^3$ Pa s µm$^4$ (left) and 0 (right). The bars represent the averages of the cycle-means of dissipations pooled from all the five sperm samples in each genotype; the error bars represent 1 standard deviation in each direction in the set of pooled cycle-means. (D) Statistical distributions of the cycle-means of powers from the WT (dark color boxes) and KO (light color boxes) samples pooled together over the entire tail (left), mid-piece (middle), and principal piece (right). The top and bottom panels are for $\eta_N = 10^3$ Pa s µm$^4$ and 0, respectively. In (C) and (D), unpaired two-tailed t-tests are used to compare population means; **** refers to a significance level of $p \leq 10^{-4}$ , *** $p \leq 10^{-3}$, ** $p \leq 10^{-3}$, * $p \leq 0.05$. Differences are not significant (n.s.) when $p > 0.05$.

The online version of this article includes the following source data and figure supplement(s) for figure 5:

**Source data 1.** Numerical data for *Figure 5A*.
**Source data 2.** Numerical data for *Figure 5B*.
**Source data 3.** Numerical data for *Figure 5C*.
**Source data 4.** Numerical data for *Figure 5D*.
**Figure supplement 1.** Comparison of population means of time-averaged dissipations obtained with the five wildtype and *Crisp2* knockout mice sperm samples obtained with (A) $\eta_N = 10^3$ Pa.s.m$^4$ and (B) $\eta_N = 0$ Pa.s.m.
**Figure supplement 2.** Comparison across genotypes of population means of time-averaged powers obtained with the five wildtype and *Crisp2* knockout mice sperm samples o $\eta_n = 10^3$ Pa.s.m$^4$ is shown in **A** and $\eta_n = 0$ Pa.s.m$^4$ in **B**.
**Figure supplement 3.** Comparison of pooled averages of the cycle-means of hydrodynamic dissipation in the tail (black) and dissipation at head due to hydrodynamic and tethering resistances (purple): $p \leq 10^{-3}$, * $p \leq 0.05$. $\eta_n = 10^3$ Pa.s.m$^4$ is shown in **A** and $\eta_n = 0$ Pa.s.m$^4$ in **B**.

negative. This would mean that, in that phase of the mean cycle, the axoneme does not drive the motion of the flagellum, but rather, the majority of the motors are being driven backward. The overall motion of the flagellum during that phase of the cycle is powered mostly by the release of the potential energy stored elastically in the body of the flagellum. This appears to be physically unrealistic. On the other hand, *Figure 5A* shows that, above $\eta_n = 10^3$, although $p^a$ has negative domains, the net instantaneous power is always positive since its minimum value in the mean cycle, $P^a_{min}$, is positive. With a value of $\eta_n > 10^3$, the motion of the flagellum is always driven by the power input from the axoneme at all times during the beat cycle.

In *Figure 5B, D*, we plot results for the energetic variables obtained with $\eta_n = 0$ and $10^3$ Pa s µm$^4$. With either value of $\eta_n$, *Figure 5B* shows that time averages, $\overline{\overline{P}}^{mi}$, of the net motor power input (*Equation 73*), calculated across all beat cycles in each sample, appear positively and linearly correlated with time averages, $\overline{\overline{P}}^{hd}$, of the hydrodynamic dissipation rate. This suggests that average hydrodynamic dissipation, which only needs the application of RFT, can be used as an indicator of the average motor input, which requires a more involved calculation. We find that the dissipation at the head against hydrodynamic and tethering forces is just a small fraction of the hydrodynamic dissipation across the tail region (*Figure 5*). Therefore, the excess of the time-averaged motor power input above the hydrodynamic dissipation is required to primarily overcome the different sources of internal dissipation in the tail.

Two different statistical approaches are possible for comparing the different kinds of dissipations within a genotypical population and for comparing the energetics across the WT and KO mice sperm. In the first approach, we can compare the population means of the time averages of samples. We recall that, for any single sperm sample, the arithmetic mean of the cycle-means of a quantity over all the cycles of that sample is the same as the time average for that sample. Within each genotype, a one-way ANOVA reveals that, for all the different energetic quantities, the time-average values of the individual sperm samples are distinctly different from the overall population mean for that genotype obtained by pooling all the cycles from the samples together ($p \ll 10^{-4}$; Appendix 2). In other words, there is significant sample-to-sample variation in the time averages of the energetic quantities. Due to the large sample-to-sample variation within each population, the standard deviations are large and although differences between the levels of the different sources of dissipation appear visually apparent, they are statistically not significant due to the small number of sperm samples (*Figure 5*). We, therefore, need to take the second approach and pool together all the individual cycles from each sample in a genotype to create a much larger set of individual time cycles for each genotype. With this approach, a clear picture emerges with statistical significance judged by

unpaired, two-tailed Student's $t$-tests ($p \ll 10^{-4}$; Appendix 2). We observe in **Figure 5C** that motor dissipation is substantial when compared with the hydrodynamic dissipation. In the WT samples, with either value of $\eta_{\mathrm{n}}$, the motor dissipation (135 fW) is clearly larger than the hydrodynamic dissipation (89.1 fW). In the KO samples, the motor dissipation (48.7 fW) is smaller than the hydrodynamic dissipation (66.3 fW), but of comparable magnitude. As discussed earlier, $\eta_{\mathrm{n}} = 10^3$ Pa s μm$^4$ is the critical value in **Figure 5A** that is required to achieve a beat pattern wherein motors deliver net positive power across the whole beat cycle. **Figure 5C** shows that, at this value of $\eta_{\mathrm{n}}$, dissipation due to internal friction (magenta) dominates above either motor (blue) or hydrodynamic dissipation (black bars) in either WT or KO samples. The data in **Figure 5C** thus leads us to conclude that, in wall-tethered WT as well as *Crisp2* KO mice sperm beating in an aqueous medium, the total internal dissipation due to motor and internal friction is considerably larger than the external hydrodynamic dissipation.

The box-plots in **Figure 5D** summarize the statistics of the entire pool of cycle-averaged powers for each genotype obtained with $\eta_{\mathrm{n}} = 10^3$ Pa s μm$^4$ (top panel) and with zero internal friction (bottom panel). We find that the net input from the dynein motors in sperm from *Crisp2* KO mice is significantly smaller than the power input in the corresponding WTs. This is observed over the entire tail. We further find that each kind of dissipation – hydrodynamic, motor, or internal friction – is smaller in sperm from *Crisp2* KO mice. These observations in **Figure 5D** are consistent with those in **Figure 4A** that the *Crisp2* KO samples have smaller beating amplitudes over the entire flagellum. The rapidity of the beating, that is, the mean beat frequency, could also be an important factor in determining the rate of energy dissipation. In the samples studied here, however, due to the large variability in cycle times, we do not find a significant difference ($p > 0.01$ in a Student's $t$-test; Appendix 2) between the population means of the cycle times (0.16 s and 0.18 s for WT and KO, respectively) or their reciprocals (7.19 Hz and 7.2 Hz, respectively) even after pooling the cycle times from the samples from each genotype together.

Further analysis of the spatial distribution of the dissipations between the mid-piece and principal piece is shown in **Figure 5D** and Table S-3. In both genotypes, the hydrodynamic dissipation occurs primarily due to the motion of the principal piece as expected. In contrast, most of the motor dissipation appears to occur in the mid-piece region in the WT population (average of 110 fW compared to 25.1 fW in the principal piece). In the KO samples, on the other hand, motor dissipation in both mid-piece and principal piece is similar (averages of 29.1 and 19.5 fW, respectively). With $\eta_{\mathrm{n}} = 10^3$ Pa s μm$^4$, the average internal dissipation in the WT population in the mid-piece (108 fW) is similar to that over the entire principal piece (131 fW). However, in the KO population, the internal dissipation in the mid-piece (32.4 fW) is much lower than in the principal piece (168 fW). This latter value is also higher than average internal dissipation in the KO samples, despite their more vigorous motion. The physical significance of this spatial distribution of the motor dissipations or the variations between the WT and KO species are not clear at this stage and require further detailed investigation.

## Discussion

In recent years, a number of studies have used image analysis of flagellar or ciliary waveforms to quantify beating patterns (**Brumley et al., 2014**; **Sartori et al., 2016**). Particle tracking (**Guasto et al., 2010**) or particle image velocimetry (**Drescher et al., 2010**) techniques have further provided a detailed picture of the dynamic velocity fields around beating filaments. These measurements have provided rich information on the nature of the beating patterns themselves (**Ma et al., 2014**; **Werner et al., 2014**; **Wan et al., 2014**) and on hydrodynamic quantities, such as the total hydrodynamic dissipation and flow features such as hydrodynamic singularities, vortices, etc. (**Ishimoto et al., 2017**; **Brumley et al., 2014**; **Gallagher et al., 2019**). Such measurements have further been used to test and refine models of axonemal dynamics (**Riedel-Kruse et al., 2007**; **Mondal et al., 2020**).

Our study contributes further to this body of work. Firstly, we have used the cycles in the phase space of POD shape coefficients to unambiguously split the data into individual time cycles. This enables the collection of data over several cycles and the calculation of mean cycles for all variables associated with flagellar beating. When used with tethered sperm, we can collect sufficient data to make statistically significant observations despite the large variability in beating patterns. Secondly,

while studies have thus far focused on external hydrodynamics and internal forces, we have shown that energy flows within sperm flagella can be extracted using standard conservation principles. The Chebyshev-POD technique proposed here provides the smooth shape modes required for the calculation of the spatial derivatives that appear in the equations. We have shown that we can use these methods to compare, in a statistically meaningful manner, the energetics of different sperm populations.

This could potentially be used to systematically explore the effect of genetic mutations on sperm energetics. Here, we have demonstrated such comparison between sperm of WT and *Crisp2* KO genotypes. The CRISPs are the sub-clade of the CAP superfamily proteins that are expressed in the male reproductive tract. *Crisp2* is further known to be incorporated internally into the sperm flagellum (*O'Bryan et al., 1998*) and is expected to act by regulating ion channels on the cell or organelle membranes (*Lim et al., 2019*). Although CRISPs are not essential for fertility (*Hu et al., 2018*; *Lim et al., 2019*; *Da Ros et al., 2008*), we see here that a lack of *Crisp2* significantly reduces the mechanical power input from the axoneme in sperm, which in turn appears to be responsible for slower beating with smaller amplitude.

Our results also reveal some fascinating new features of flagellar energetics that appear to be shared by all of our samples. We firstly see that along the filament there exist distinct phases during each cycle where dynein motors in the axoneme are driven back against the torques they exert by the motion of the rest of the flagellar body. It is known that dynein motors are regulated to create a traveling wave of forces, and hence turning moments, that propagates down the flagellum (*Lin and Nicastro, 2018*). Since the active power density $p^a = m^a \omega$, the periodic occurrence of positive and negative domains in the active power distribution in *Figure 3D* shows that at any location along the tail $m^a$ and $\omega$ are in the same direction (i.e., of the same sign) in some parts of a beat cycle and in opposite directions (i.e., of opposite sign) in other parts of the cycle. In other words, the rotational velocity and the moment exerted by the dyneins are out of phase with one another, as shown in *Figure 6*.

This is in line with current thinking on axonemal dynamics. Several ideas have been presented in the past for the generation of the beating patterns by the axoneme. In a landmark study, *Riedel-Kruse et al., 2007* compared the predictions of many of these with experimental observations of planar beating in bull sperm that were either head-tethered or swimming freely in circles for long adjacent to a glass-slide wall. It was shown that the best agreement with experiments is obtained with the sliding-control model of Jülicher and co-workers (*Camalet et al., 1999*; *Camalet and Jülicher, 2000*). In this model (in the notation of the current paper), the active moment is related to

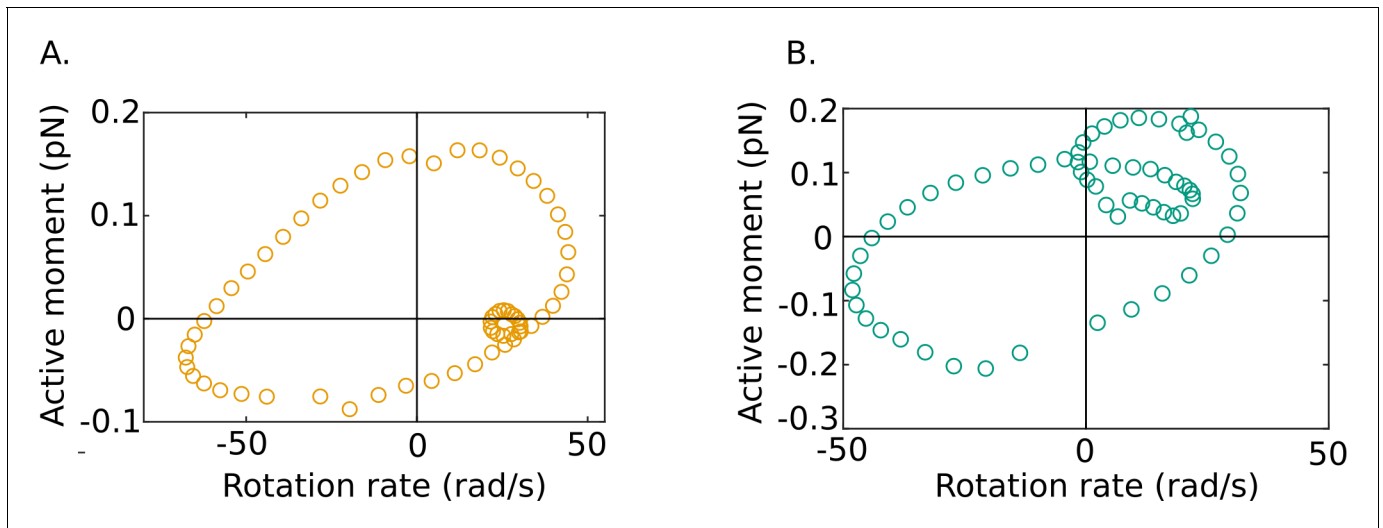

**Figure 6.** Out-of-phase mean beat cycles of active moment density and angular rotation rate at $s = 0.5$ in (A) wildtype (WT)-1 and (B) knockout (KO)-1 samples.
The online version of this article includes the following source data for figure 6:

**Source data 1.** Numerical data for *Figure 6*.

the local internal shear and shear rate through an equation of the form $m^{\mathrm{a}} = K\gamma + \lambda\, \partial\gamma/\partial t$, where $\gamma$ is the local shear strain. In the parlance of control theory, this model proposes that motors are regulated by the location deformation through a mechanism that follows a proportional-derivative control logic. More recently, *Mondal et al., 2020* suggested a variant with proportional-integral control logic instead, that is, where $m^{\mathrm{a}} + \beta\, \partial m^{\mathrm{a}}/\partial t = K\gamma$. In either case, when the equation for regulation of the active moment is coupled with the equations for the rest of the passive material of the flagellum, an oscillatory instability emerges in certain ranges of the controller constants. This triggers a traveling wave that propagates down the filament, leading to beating patterns that are similar to those observed experimentally . It is further found with these models that the controller constants to achieve oscillations are negative, indicating that the active moment exerted by the dynein motors is down-regulated by the load exerted back on the motors due to the local shear deformation in the filament and its time rate of change. This also appears to be consistent with the recent experimental finding that dynein motors are always primed to deliver forces on microtubules but are inhibited when a curvature wave passes through their location (*Lin and Nicastro, 2018*).

It is possible that regulation of $m^{\mathrm{a}}$ could more generally be described by an equation of the form, $m^{\mathrm{a}} + \beta\, \partial m^{\mathrm{a}}/\partial t = K\gamma + \lambda\, \partial\gamma/\partial t$, which corresponds to proportional-integral-derivative (PID) control. Such regulation of $m^{\mathrm{a}}$ immediately means that, when stable traveling waves are generated, the local rotation rate, $\omega$, (which is proportional to $\partial\gamma/\partial t$) will be systematically out of phase with $m^{\mathrm{a}}$, as is indeed observed in *Figure 6*. There will necessarily, therefore, be phases in each cycle when the two variables will be of opposite sign and $p^{\mathrm{a}} = m^{\mathrm{a}}\omega$ will always be negative in those phases.

The mechanical work done back on the motors during such phases by the passive elements of the filament must be quickly dissipated in some form since the motors cannot store the energy that is received nor reconvert it back to ATP. What, then, is the internal mechanism behind this additional dissipation? Riedel-Kruse et al. pointed out that the sliding-control model had to allow for relative sliding between microtubules at the basal end to obtain experimental agreement and that frictional resistance to basal shearing is important for the model to predict stable oscillations. Mondal et al. analyzed axonemes isolated by demembranating Chlamydomonas cilia and found that external hydrodynamic friction is too small to explain the stable beating pattern observed. They then showed that their sliding-control model predicts stable oscillations when coupled with equations that include passive filament elasticity and internal frictional resistance to the *shear* deformation rate. These sources of internal friction are not modeled in the present study, where we have treated the flagellum as an *unshearable* Kirchhoff rod. As *Figure 5A* shows, we find that, if internal friction is absent or insufficient, then the observed motion would mean that, for a significant duration of the mean cycle, the filament may as a whole be driving the motors backward. While this unphysical picture is eliminated when a sufficiently high internal friction coefficient is used, we still observe motor dissipation due to $m^{\mathrm{a}}$ and $\omega$ being out of phase with one another.

The key point is that, while some or all of these different frictional contributions may be necessary for an internally driven filament to oscillate stably, if the local regulation of the active moment in general follows PID logic, then the out-of-phase moment and local deformation rate will lead to phases of negative active power, irrespective of the nature of internal or external friction. This points to the existence of a separate dissipative mechanism associated with the dynein motors themselves. There is already evidence that dyneins can dissipate energy locally. It is known that dynein motors can cycle through conformational changes driven by ATP binding and hydrolysis even when not driving microtubule sliding (*Kon et al., 2005*). Optical-tweezer experiments on dyneins bound to static microtubules have further shown that dyneins can steadily be driven in the reverse along the microtubule by an external load by forces larger than the stall force for these motors (*Gennerich et al., 2007*). The force required is more than that required to move unbound motors at the same velocity. This work done to drive the motors backward must be dissipated locally by a mechanism other than just the hydrodynamic frictional resistance of the motors to motion. Our results show that such motor dissipation can be a large part of the energy budget within the flagellum.

As pointed out above, our results in *Figure 5* indicate that bending friction in the accessory structures surrounding the axoneme could also be significant. While most current models of flagella or cilia assume that the flagellum is a purely elastic filament, it is beginning to be recognized that internal friction plays an important role in flagella and cilia (*Riedel-Kruse et al., 2007*; *Mondal et al., 2020*; *Klindt et al., 2016*). An internal friction coefficient of $\eta_{\mathrm{n}} = 10^{3}$ Pa s μm$^{4}$ is the minimum required to obtain physically realistic axonemal power input. It is possible that the internal coefficient

is larger than this value. The ratio $\eta_n/\kappa_n$ represents a characteristic internal viscoelastic time scale for the passive flagellar material. If internal friction dominates the dynamics, we should expect to see the observed mean frequency of a beat cycle, $\bar{f} \sim \kappa_n/\eta_n$. The observed beat frequency of 7 Hz and $\kappa_n = 7 \times 10^4$ Pa $\mu m^4$ suggests $\eta_n \sim 10^4$ Pa s $\mu m^4$. Systematic measurements of the bending and other internal friction coefficients in flagella and cilia through single-cell microrheological techniques are therefore essential for a better understanding of their dynamics.

Our experiments were conducted with an aqueous buffer with cells beating close to a wall. It is natural to ask, therefore, how the results here would change with either medium viscosity or in the absence of the greater and more anisotropic hydrodynamic resistance due to wall. If the kinematics of the beating pattern remain unchanged, changes in medium viscosity or the distance from the wall would trivially result in changes in the magnitude of the hydrodynamic friction coefficients, and pro-portional changes in the contribution of hydrodynamic dissipation. However, the response to changes in the viscous resistance may be considerably more complex. It is known that the beating pattern changes dramatically with an increase in medium viscosity (*Smith et al., 2009*; *Kirkman-Brown and Smith, 2011*). Well away from a wall, the beating is non-planar, with helical traveling waves, and as sperm approach a wall, the beating becomes planar and cells appear to 'slither' quickly across the surface (*Nosrati et al., 2015*). Although the mechanisms behind these qualitative changes in beating waveforms are still unknown, it is likely that they are the result of the strong cou-pling of the motor regulation and the viscoelasto-hydrodynamics of the passive filament. With such changes in the waveform, the internal frictional and motor dissipation can also be expected to change appreciably. A related question pertains to the effect of the tethering constraint at the head. The constraint results in an additional force and torque being imposed at the head. Removing the constraint will alter the external loading on the cell and may result in a qualitatively different beating pattern and energetics. We nonetheless expect that, even in freely swimming sperm, motor dissipa-tion and internal friction will be important.

Moreover, our observations of the effect of the *Crisp2* mutation on the waveform and energetics are also likely to be independent of the effect of the tethering. The smaller beating amplitudes result in a smaller motor dissipation in the KO samples that is similar in magnitude with the smaller hydro-dynamic dissipation in those samples, whereas in the WT samples, the motor dissipation is clearly larger than the hydrodynamic dissipation. The approach presented here can similarly be used to sys-tematically explore the role played by other proteins and signaling agents on the internal dynamics and energetics of flagellar beating.

# Materials and methods

**Key resources table**

| Reagent type (species) or resource | Designation | Source or reference | Identifiers | Additional information |
|---|---|---|---|---|
| Gene (*Mus musculus*) | *Crisp2* | *Lim et al., 2019* | - | - |
| Strain, strain background (*Mus musculus*) | C57BL/6N | *Lim et al., 2019* | PMID:30759213 | Mice produced through the Australian Phenomics Network |
| Biological sample (*Mus musculus*) | Sperm | *Lim et al., 2019* | - | Collected from the cauda epididymis and vas deferens using the backflushing method |
| Chemical compound, drug | TYH medium with 0.3 mg/ml BSA | *Lim et al., 2019* | - | Buffer media for sperm |
| Software, algorithm | MATLAB, MATLAB Image Processing Toolbox, Fiji | | SCR 001622,SCR 002285 | Code (*Nandagiri, 2021*) and original videos (*Nandagiri et al., 2020*) available for public access |

## Sperm sample preparation

Generation of KO mouse models and all animal procedures were approved by the Monash University Animal Experimentation Ethics Committee. The mouse KO line were maintained on a C57/BL6N

background. Sperm were collected from *cauda epididymis* and *vas deferens* using the back-flushing method (*Lim et al., 2019*) in modified TYH medium (135 mM NaCl, 4.8 mM KCl, 2 mM CaCl$_2$, 1.2 mM KH$_2$PO$_4$, 1 mM MgSO$_4$, 5.6 mM glucose, 0.5 mM Na-pyruvate, 10 mM L-lactate, 10 mM HEPES, pH 7.4). The samples were stored in dark at 37 ˚C until imaging. Sperm samples WT-1 and -2 were from the same individual, WT-3 and -4 were from another individual, and WT-5 was from a third individual mouse. All the five KO samples were from separate individuals.

## Tethering and imaging

Sperm motility was investigated in a custom-made observation chamber. Briefly, two strips of double-sided tape (90 µm nominal thickness) were affixed to a glass slide 16 mm apart. A drop of 40 µl of sperm suspension was placed between the two strips and sealed against evaporation with 17 mm square coverslips (Thermo Fisher Scientific, No. 1.5).

Mouse sperm have flat falciform (hook-shaped) heads. A detailed study by *Woolley, 2003* showed that freely swimming mouse sperm are hydrodynamically drawn to walls and mostly stabilize with the left sides of their flat heads held against the surface. It was also found that the plane of the left side of the flat head makes an angle less than 180˚ with the flagellum at the neck. This enables sperm following the left-side rule to stabilize to beating in a plane parallel to the wall.

We have taken advantage of this nearly planar beating close to walls to design our experiments. In our experiments, the TYH medium was supplemented with 0.3 mg/ml of BSA, which causes sperm swimming at the wall to adhere to the glass slide at the bottom of the imaging chamber. The out-of-plane excursions in the resolved portion of the tail appear limited to less than 2 µm (Appendix 2). This beating is clearly resolvable within the depth of field of the microscope. Sperm tethered at their heads with flagella beating freely within the focal plane were chosen for video imaging and subsequent analysis. Imaging is done from above the sperm cell.

An Olympus AX-70 upright microscope equipped with a U-DFA 18 mm internal diameter darkfield annulus, an 20 × 0.7 NA objective (UPlanAPO, Olympus, Japan), and incandescent illumination served as the platform for the imaging system. All extraneous optical elements were removed from the detection light paths to maximize system light efficiency. An ORCA-Flash4.0 v2+ (C11440-22CU) sCMOS camera (Hamamatsu, Japan) was used for capturing images. This system leverages a high frame rate for motion capture, an exceptional 82% QE for the low level of light and the small 6.5 µm pixel size to increase system spatial resolution (*Stuurman and Vale, 2016*; *Beier and Ibey, 2014*; *Saurabh et al., 2012*).

The optical lateral resolution was 0.479 µm at a reference wavelength of 550 nm. With the 6.5 µm pixel size of the ORCA sCMOS and the system magnification factor of 20, the best-case lateral resolution of 0.650 µm (0.325 µm/pixel) at the Nyquist–Shannon sampling was sufficient to spatially resolve the tip of the sperm tail. A 512 × 512 pixel region of interest therefore corresponded to an experimental sample FOV of 166.4 × 166.4 µm, which was sufficient for most of the experiments reported here. Occasionally, sperm with stiffer flagella required an FOV increase with a reduction of approximately 0.8 frames per second (fps) for each pixel increase.

Image data was free-streamed to a Xeon E5-2667 computer (with a 12-core CPU running at 2.9 GHz supplemented by 64 GB of DDR3 RAM and 1 TB SSD hard drive in a RAID0 configuration) via a dedicated Firebird PCIe3 bus 1xCLD Camera Link frame-grabber card (Active Silicon, UK) at the 8.389 MB/s memory buffer speed of the camera. This resulted in a capture frame rate of approximately 400 fps. The best-case blur-free motion capture of the system at this frame rate corresponds to element point velocities of 130 µm/s. The Fiji image-processing package was used for image capture control along with the Micro-Manger Studio plugin (version 1.4.23) for multidimensional acquisition (*Beier and Ibey, 2014*) set to 4000 time points, zero time point interval, a 2.0 ms exposure time. The data was written as an image stack.

Camera resolution can be increased to exceed optical resolution by replacing the 180 mm tube lens with a 250 mm tube lens. The region of interest would then increase to 714 × 714 pixels, with the capture frame rate being reduced to approximately 286 fps. Frame exposure can be likewise increased to 3.25 ms to allow for a superior signal-to-noise ratio.

### Image analysis and skeletonization

The videos of the sperm samples are available for public access (*Nandagiri et al., 2020*). The mean of the grayscale intensity at each pixel location across all the frames was used to construct a background image. This was then subtracted from each frame to remove the background. The contrast was then adjusted to enhance the foreground grayscale intensity. Median filters of different sizes were applied to remove noise. The grayscale image was then smoothened with a Gaussian filter before binarization at a threshold computed by Otsu's method (*Otsu, 1979*). Connected components in the binarized image were then located and classified according to size and eccentricity. The sperm body is expected to have the largest size among the objects in the frame. An oval (i.e., an ellipse) is fitted around each body. The eccentricity is a measure of the deviation of the oval from a perfect circle. An oval fitted around the whole sperm body will be highly elongated and will have a high eccentricity. These two criteria were used to automatically identify the sperm body in each frame and remove other extraneous objects. Morphological thinning was then applied to the segmented image to extract a skeleton of the sperm tail. Spurious branches on the skeleton were automatically identified and removed to give an unbranched skeleton. The skeleton at this stage is rough, with noisy burrs that are then smoothed out using low-pass filtering. The resulting smoothed curve representing the sperm body is henceforth referred to as the centerline (*Figure 7*). Since the algorithm treats each frame independently of all others, frames were processed on separate processors on a high-performance computational cluster.

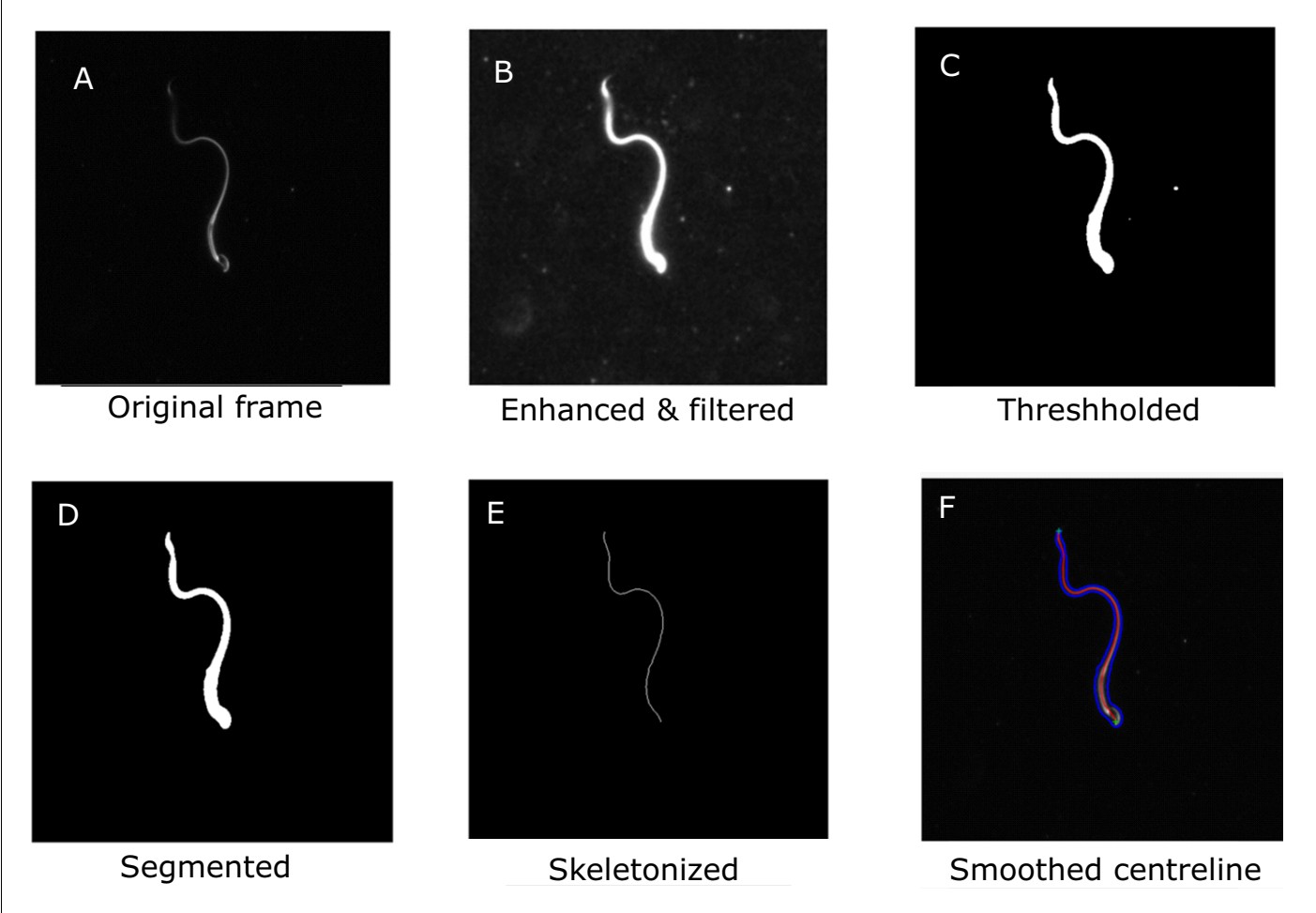

**Figure 7.** Main steps in the image-processing algorithm shown for a single frame. A. The original frame B. Enhanced and filtered C. Thresholded frame D. Segmented E. Skeletonized frame F. Smoothed centreline.

The arc length between each adjacent pair of points was calculated and the overall contour length of the centerline in each frame was obtained. Motion of the sperm body out of the plane of focus leads to blurring and loss of contrast and intensity of the image, which in turn increases errors in the automated processing of the images. This is particularly problematic at the tail end of the flagellum. As a result, the skeleton obtained is truncated at the tail end, resulting in a loss of total contour length of the captured skeleton. Videos with significant loss of length were discarded, and only videos showing largely in-plane beating, with deviations smaller than 10% from the mean contour length, were considered for further analysis. In each of the samples selected for further analysis, the maximum contour length across all the video frames is taken to be the cell body length, $L$.

For each video, the time-averaged end-to-end straight line was first determined. Sperm centerlines in every frame were rotated by an angle to align this line with the horizontal $x$-axis. The centerlines in a video were reflected about the horizontal axis if necessary to orient the head-hook concave downwards in all videos. At this stage, the pixel points on the centerline were not uniformly distributed along the length of the sperm body. That is, the arc length between each adjacent pair of points is not the same along the centerline. The $x$ and $y$ coordinates for each centerline point were linearly interpolated to obtain a large number of points (~200) distributed uniformly with the same difference in the arc length $s$ between adjacent points. Frames were also not always equally spaced in time since poor-quality frames were discarded. Linear interpolation in time was applied across the two frames on either side of a missing frame to compute the centerline in the missing frame. Tangent angles to the horizontal were computed at each $s$ in every frame. A Butterworth low-pass filter was used to spatially smooth the tangent-angle-versus-$s$ data in each frame. The values of $s$ in each frame are normalized by the body length, $L$.

## Data processing

The head and imaged-tail regions are defined as $s \in [0, s_{\mathrm{N}}]$ and $s \in [s_{\mathrm{n}}, s_{\mathrm{t}}]$, where $s_{\mathrm{n}} = 0.1L$, and $s_{\mathrm{T}}$ is the maximum value of $s$ for which pixel data is available for every time sample (typically, $s_{\mathrm{T}} = 0.85\,L$). Since the data for $s > s_{\mathrm{T}}$ is not available at all time steps, this data is neglected.

When working with Chebyshev polynomials in the tail region, we define a rescaled variable that maps the domain $[s_{\mathrm{N}}, s_{\mathrm{T}}]$ onto $[-1, 1]$:

$$\xi = 2\left(\frac{s - s_{\mathrm{N}}}{s_{\mathrm{T}} - s_{\mathrm{N}}}\right) - 1\,. \tag{25}$$

The inner product of a pair of functions $f$ and $g$ with respect to the Chebyshev weighting function $w(\xi)\,, = 1/\sqrt{1 - \xi^2}$ is defined as $(f, g) = \int_{-1}^{1} f(\xi)\, g(\xi)\, w(\xi)\, d\xi$. The norm of $f$, $\|f\| = \sqrt{(f, f)}$. We further denote time averages, $t_{max}^{-1} \int_{0}^{t_{\max}} \ldots dt$ as $\langle \ldots \rangle$.

There are four stages in calculating flagella energetics, starting from the raw tangent-angle data obtained from the centerlines after image processing. This original tangent-angle function is denoted as $\hat{\psi}$.

1. In this stage, the intermediate tangent-angle profile, $\widetilde{\psi}$, is determined from $\hat{\psi}$. Firstly, at each time instant, the raw centerline data in the head region is analyzed to fit the tangent-angle profile corresponding to a rigid body motion that rotates about the point of tether. This gives the motion of the neck junction at the end of the head region. A 20th-order Chebyshev polynomial is then fitted through the pixel data in the tail region and is also $C^2$-continuous with the rigid-body motion of the head. The tangent-angle profile combining the rigid-body fit at the head and the Chebyshev polynomial through the tail is $\widetilde{\psi}$.
2. The next stage is to perform C-POD to obtain the optimally compact representation, $\psi(s, t)$, of the tail region in the form shown in *Equation (17)*.
3. The C-POD representation, $\psi(s, t)$, is then used to calculate other geometric, kinematic, and dynamic quantities.
4. Mean cycles of all physical quantities and the standard errors in the means are then calculated.

Each of these stages is described further below.

## Intermediate tangent-angle profile

The raw centerline data $\widehat{\psi}$ in the head region does not satisfy the rigid-body motion conditions since the large and diffuse image of the head leads to errors during skeletonization in identifying its centerline consistently. The sufficient condition that the head region rotates about a single point as a rigid body is that $\partial C/\partial t = \partial \omega/\partial s = 0$. To impose this, the time-averaged tangent-angle profile $\widetilde{\psi}_0(s) = \langle \widehat{\psi}(s,t) \rangle$ is first calculated from the raw data in this domain. Then, the tangent-angle profile in this domain is set to the following to ensure the rigid-body conditions:

$$\widetilde{\psi}(s,t) = \widetilde{\psi}_0(s) + \widetilde{B}_0(t), \tag{26}$$

where

$$\widetilde{B}_0(t) = \widehat{\psi}(s_\mathrm{n},t) - \widetilde{\psi}(s_\mathrm{n}). \tag{27}$$

The time average of $B_0$ is thus zero. The rotation rate, $\omega = \partial \widetilde{\psi}/\partial t = \widetilde{B}_0/dt$, is uniform and non-zero for the whole head region. With this profile, the tangent values at the neck are given by $\psi_\mathrm{n}(t) = \widetilde{\psi}_0(s_\mathrm{n}) + \widetilde{B}_0(t)$. The time-independent $s$-derivatives, $\psi'_\mathrm{n}$ and $\psi''_\mathrm{n}$ are determined from the values of $\widetilde{\psi}_0$ adjacent to the neck in second-order backward-difference formulae.

A Chebyshev polynomial,

$$\widetilde{\psi}(\xi,t) = \sum_{k=0}^{P} a_k(t)\, T_k(\xi), \tag{28}$$

of order $P = 20$ is fitted to the data in the imaged tail region at each time, $t$. Here, $T_k$ is the $k$ th Chebyshev polynomial of the first kind (*Hildebrand, 1987*). The fitted Chebyshev polynomial must also satisfy boundary conditions at $\xi = -1$ so that it is $C^2$-continuous with the tangent profile of the rigid head region across the neck. No boundary conditions are imposed at the other boundary at $\xi = 1$ since that end of the imaged region is not the physical end of the tail. (The physical boundary conditions at the tail tip are accounted for through the energy balance [*Equation 73*], as discussed earlier.)

To ensure $C^2$-continuity across the neck, we must have at $\xi = -1$,

$$\widetilde{\psi}(-1,t) = \widetilde{\psi}_\mathrm{n}(t); \quad \left.\frac{\partial \widetilde{\psi}}{\partial \xi}\right|_{\xi=-1} = \frac{(s_\mathrm{t} - s_\mathrm{n})}{2} \widetilde{\psi}'_\mathrm{n}(t); \quad \left.\frac{\partial^2 \widetilde{\psi}}{\partial \xi^2}\right|_{\xi=-1} = \left(\frac{s_\mathrm{t} - s_\mathrm{n}}{2}\right)^2 \widetilde{\psi}''_\mathrm{n}(t), \tag{29}$$

where $\widetilde{\psi}_\mathrm{n}$, $\widetilde{\psi}'_\mathrm{n}$, and $\widetilde{\psi}''_\mathrm{n}$ are the values of the tangent angle and its first two $s$-derivatives at the neck, respectively. These values at the neck are determined from the motion of the rigid head, as discussed above.

We determine the set of coefficients $a_k$ as those that minimize $S = \|\widehat{\psi} - \widetilde{\psi}\|^2$, the mean square error between the raw data, $\widehat{\psi}$, and the Chebyshev polynomial, $\widetilde{\psi}$, while also satisfying the boundary conditions at $\xi = -1$ in *Equation (29)*. Using the properties of Chebyshev polynomials and Lagrange's method of undetermined coefficients, and using standard methods and Gaussian quadrature to approximate integrals, we obtain

$$a_k(t) = a_k^\star(t) + \frac{K_1}{2\gamma_k}(-1)^{k+1} + \frac{K_2}{2\gamma_k}(-1)^k k^2 + \frac{K_3}{2\gamma_k}(-1)^{k+1}\left(\frac{k^4 - k^3}{3}\right), \tag{30}$$

where $\gamma_k = (1 + \delta_{0,k})/(2(P+1))$, $\delta_{i,j}$ is the Kronecker $\delta$-function, $K_1$, $K_2$, and $K_3$ are the Lagrange multipliers. The $k$ th unconstrained Chebyshev coefficient,

$$a_k^\star(t) = \frac{1}{\gamma_k} \sum_{i=0}^{P} \widehat{\psi}(\xi_i,t)\, T_k(\xi_i), \tag{31}$$

where $\xi_i$ is the $i$ th root of the $P+1$ th Chebyshev polynomial (*Hildebrand, 1987*). The values of $T_k(\xi_i)$ can be calculated using standard recursion relations (*Hildebrand, 1987*). Substituting from *Equation (30)* in the boundary conditions into *Equation (29)* results in a system of linear equations

that can be solved for the Lagrange multipliers. Inserting these values back into *Equation (30)* gives the Chebyshev coefficients in the imaged tail region. The resulting $\widetilde{\psi}$ is consistent with boundary conditions at the neck.

## C-POD of the tail region

The C-POD provides advantages over the 'empirical' POD used previously for sperm (*Werner et al., 2014*; *Ma et al., 2014*). The empirical POD is applied directly on the discrete data to produce shape modes that are numeric vectors. The discrete nature of the modes makes high-order spatial derivatives computed from them susceptible to noise. The C-POD approach here allows derivatives to be computed without noisy artifacts. Further, specific restrictions on the shape at the boundaries can be conveniently imposed.

We first recall key aspects of the general POD technique to obtain the optimal mutually orthogonal basis functions. The Chebyshev polynomials $T_k$ themselves constitute a set of mutually orthogonal basis functions. At any $t$, $\widetilde{\psi}$ is a polynomial of order $P$ that is expanded in terms of $P+1$ Chebyshev polynomials. Given a small number $M<P+1$, say $M=2$, any linear combination of $M$ of the Chebyshev polynomials can be expected to be a poor approximation of the full $P$th-order polynomial, $\widetilde{\psi}$. The technique of POD allows us to find a set, $\{\psi_m\}$, of $M$ unique orthogonal functions different from $T_k$ such that a linear combination of these provides the *best* approximation of $\widetilde{\psi}$ possible, given the choice of $M$. The gain is that we need to track only the set of $M$ coefficients $\{B_m\}$ as functions of time rather than the larger set of all the $P+1$ time-dependent Chebyshev coefficients, $\{a_k\}$.

The time-averaged profile in the imaged-tail region is $\psi_0(\xi) = \langle\widetilde{\psi}(\xi,t)\rangle$. The deviation of the original function $\widetilde{\psi}(\xi,t)$ from this time average is

$$\widetilde{\chi}(\xi,t) = \widetilde{\psi}(\xi,t) - \widetilde{\psi}_0(\xi), \tag{32}$$

and the spatial two-point cross-correlation of $\widetilde{\chi}$ is

$$C(\xi,\mu) = \langle\widetilde{\chi}(\xi,t)\,\widetilde{\chi}(\mu,t)\rangle. \tag{33}$$

It can be shown that the set of optimal basis functions for the POD are the eigenfunctions of this two-point cross-correlation (*Lumley, 1967*; *Holmes et al., 2012*). That is, an optimal shape mode, $\psi_m$, is such that

$$(C(\xi,\mu),\psi_m(\mu)) = \lambda_m\,\psi_m(\xi), \tag{34}$$

where $\lambda_m>0$ is the corresponding eigenvalue. These eigenfunctions are mutually orthogonal, that is, $(\psi_m,\psi_n)=\delta_{m,n}$. The coefficient of the $m$th shape mode is then obtained by projection as

$$B_m(t) = (\widetilde{\chi}(\xi,t),\psi_m(\xi,t)). \tag{35}$$

These coefficients are themselves orthogonal in time, that is, $\langle B_m(t)\,B_n(t)\rangle=\delta_{m,n}\lambda_m$. The matrix algorithm for obtaining the time-independent Chebyshev coefficients of the shape modes is as follows. The Chebyshev polynomials are first normalized as follows:

$$\tau_m(\xi) = \frac{1}{\sqrt{\gamma_m}}\,T_m(\xi) \tag{36}$$

so that the inner product (with the Chebyshev weighting function) $(\tau_m,\tau_n)=\delta_{m,n}$. The Chebyshev coefficients $a_k$ of $\widetilde{\psi}$ are correspondingly rescaled as $\alpha_k=\sqrt{\gamma_k}a_k$, so that $\widetilde{\psi}(t,\xi)=\sum_{k=0}^{P}\alpha_k(t)\,\tau_k(\xi)$. The Chebyshev coefficients of the time-averaged tangent-angle profile, $\psi_0(\xi)$, and the deviation from the mean, $\widetilde{\chi}$, are then $\langle\alpha_k(t)\rangle$ and $\Delta\alpha_k(t)=\alpha_k(t)-\langle\alpha_k(t)\rangle$, respectively. From *Equation (34)*, the cross-correlation, $C(\xi,\mu)=\sum_{l=0}^{P}\sum_{k=0}^{P}\tau_k(\xi)A_{kl}\tau_l(\mu)$, where

$$A_{kl} = \langle\Delta\alpha_k(t)\,\Delta\alpha_l(t)\rangle. \tag{37}$$

The symmetric matrix **A** composed of $A_{kl}$ is equivalent to the cross-correlation matrix.

Diagonalizing the matrix $\mathbf{A} = \mathbf{V} \cdot \Lambda \cdot \mathbf{V}^{\mathrm{T}}$ yields the $P+1$ eigenvalues, $\{\lambda_m\}$, of the correlation operator as the diagonal elements of the matrix, $\Lambda$. The $m$ th column of $\mathbf{V}$ is the $m$ th eigenvector of $\mathbf{A}$. Its elements are the Chebyshev coefficients of the $m$ th shape mode:

$$\psi_m(\xi) = \sum_{k=0}^{P} V_{km} \, \tau_k(\xi) \,. \tag{38}$$

The corresponding shape coefficient can be obtained from the equation above and from *Equation (35)* as

$$B_m(t) = \sum_{k=0}^{P} \Delta\alpha_k(t) \, V_{km}. \tag{39}$$

With $\psi_m$, and $B_m$ thus determined from the original cross-correlation of $\widetilde{\chi}$, we can obtain the C-POD approximation, $\psi$, given by *Equation (17)* for any choice of $M \leq P+1$. The deviation of the C-POD approximation from the mean,

$$\chi = \psi - \psi_0 = \sum_{m=1}^{M} B_m(t)\psi_m(\xi) \,, \tag{40}$$

is an approximation of the original $\widetilde{\chi}$. The approximation improves with increasing $M$ and when $M = P+1$, $\chi = \widetilde{\chi}$ exactly, since the full set of $P+1$ eigenfunctions $\{\psi_m\}$ spans the same function space that is spanned by the set of $P+1$ Chebyshev polynomials, $\{T_k\}$. Further, using the orthogonality of the shape modes, it can be shown that

$$\langle \|\widetilde{\chi}\|^2 \rangle = \sum_{m=1}^{P+1} \lambda_m \,; \quad \langle \|\chi\|^2 \rangle = \sum_{m=1}^{M} \lambda_m \,. \tag{41}$$

Therefore, the mean-squared error in the approximation when $M < P+1$,

$$\langle \|\psi - \widetilde{\psi}\|^2 \rangle = \langle \|\chi - \widetilde{\chi}\|^2 \rangle = \sum_{m=1}^{P+1} \lambda_m - \sum_{m=1}^{M} \lambda_m = \langle \|\widetilde{\chi}\|^2 \rangle - \langle \|\chi\|^2 \rangle \,. \tag{42}$$

We can, therefore, use the ratio of the cumulative sum of the eigenvalues for any $M$, normalized by the sum of all the $P+1$ eigenvalues,

$$\Gamma_M = \frac{\sum_{m=1}^{M} \lambda_m}{\sum_{m=1}^{P+1} \lambda_m} = 1 - \frac{\langle \|\chi - \widetilde{\chi}\|^2 \rangle}{\langle \|\widetilde{\chi}\|^2 \rangle} \,. \tag{43}$$

as a measure of the accuracy of the $M$ th order C-POD representation: the closer $\Gamma_M$ is to 1, the better $\psi$ captures $\widetilde{\psi}$. As discussed earlier, $\widetilde{\psi}$ is constructed to be consistent with the neck boundary conditions (in *Equation 29*) at all times. The C-POD basis functions, $\psi_m(\xi)$, that span this function space, therefore, also satisfy the same neck boundary conditions.

## Calculation of flagellar kinematics and dynamics

The equations in the section on The soft, internally driven Kirchhoff rod model are used to calculate the active power distribution in the following manner:

1. The centerline coordinates are obtained from the tangent angle $\psi(s,t)$ by

$$x(s,t) = x_{\mathrm{h}}(t) + \int_0^s \cos(\psi(s',t)) \, ds' \,; \quad y(s,t) = y_{\mathrm{h}}(t) + \int_0^s \sin(\psi(s',t)) \, ds' \,; \tag{44}$$

where $x_{\mathrm{h}}$ and $y_{\mathrm{h}}$ are the experimentally determined coordinates of the tip of the head at any time $t$. Further, $(t_x, t_y) = (\cos\psi, \sin\psi)$ and $(n_x, n_y) = (-\sin\psi, \cos\psi)$.

2. Since the shape modes are given by *Equation (38)*, their spatial derivatives are calculated by applying standard recursion relations for the Chebyshev polynomials (*Hildebrand, 1987*). The time rates of the shape coefficients, $\dot{B}_m = dB_m/dt$, are calculated numerically using central-difference formulae.

3. We then calculate the spatial derivatives of the C-POD approximant, $\psi$, and obtain the curvature, $C = \partial\psi/\partial s$, and its derivatives. Its time derivative, $\partial C/\partial t = \sum_{m=1}^{M} \dot{B}_m \psi'_m$ and the centerline angular velocity, $\omega = \partial\psi/\partial t = \sum_{m=1}^{M} \dot{B}_m \psi_m$ are calculated.

4. Noting that,

$$\frac{\partial v_x}{\partial s} = -\omega t_y = -\omega\sin\psi\,; \quad \frac{\partial v_y}{\partial s} = \omega t_x = \omega\cos\psi, \tag{45}$$

flagellar velocities are calculated from the rotation rate as follows:

$$v_x = -\int_{s_e}^{s} \omega\sin\psi\,ds'\,; \quad v_y = \int_{s_e}^{s} \omega\cos\psi\,ds'\,. \tag{46}$$

where $s_e$ is the experimentally determined location of the tether point. The tangential and normal components of the centerline velocity, $v_t = \mathbf{v}\cdot\mathbf{t}$ and $v_n = \mathbf{v}\cdot\mathbf{n}$, are then calculated.

5. The hydrodynamic force distribution, $\mathbf{f}^h$, is calculated using *Equation (10)* and the expressions for the tangential and normal friction coefficients.

6. After $\mathbf{f}^h$, the densities, $\dot{\epsilon}$, $p^{hd}$, $p^{id}$, and $p^s$, are determined as described in the section on Dynamics and energetics from measured kinematics. The energy balance (*Equation 71*) is then used to calculate the active power density, $p^a$, across the tail region. The contribution of the non-imaged end of the tail ( $s > s_t$) to the energetics is neglected. In addition, the instantaneous power exerted on the head, $P_h^d$, is calculated using *Equation (66)*, as are the instantaneous rate, $\dot{E}$, and the powers, $P^{hd}$ and $P^{id}$, and the instantaneous motor dissipation, $P^{md}$ (*Equation 21*). Kymographs are generated using the colormaps as discussed in *Auton, 2020*.

7. The cycle-means, $\overline{P}^{hd}$, $\overline{P}^{id}$, $\overline{P}^{md}$, and $\overline{P}_h^d$, are obtained in each beat cycle by integrating the corresponding time-dependent power over that beat cycle and normalizing by its time period. The cycle-mean motor power input, $\overline{P}^{mi}$, is then calculated using *Equation (24)*.

## Mean beat cycles

The time-dependent coefficients of the dominant shape modes, $B_1$ and $B_2$, are plotted against one another. Individual beat cycles are identified from the times at which the polar angle of a point in this $B_1$–$B_2$ space is zero. In other words, a beat cycle starts when the flagellar shape is a scaled version of the first shape mode, $\psi_1$. The time phase within the $i$th beat cycle is then calculated as

$$\tau = \frac{(t - t_i^0)}{T_i}\,, \tag{47}$$

where $t_i^0$ is the starting time of the $i$th cycle and $T_i = t_{i+1}^0 - t_i^0$ is the time period of that cycle.

Functions such as $p^a(s,t)$ and $P^a(t)$ are split into individual beat cycles and, in each cycle, expressed as functions of the time phase, $\tau$. The mean of that function over the set of its cycles is computed at each $\tau$, as is the SEM. Between 40 and 60 beat cycles were captured for each sperm sample. These beat cycles are used for statistical analysis either for each sample, or for each genotypical population, as required. The mean beat cycles of beating patterns and their shaded error bands in *Figure 4B* have been obtained in this manner and by applying the graphing tools provided in *Campbell, 2020*. The averaged powers, $\overline{\overline{P}}_h^d$, $\overline{\overline{P}}^{hd}$, $\overline{\overline{P}}^{id}$, $\overline{\overline{P}}^{md}$, and $\overline{\overline{P}}^{mi}$, are computed as the averages of the corresponding cycle-means over all beat cycles in either the set of samples or the set of pooled cycles, as required.

## Acknowledgements

This research was funded by the Australian Research Council (DP190100343, DP200100659), an Interdisciplinary Research Grant from the Provost's office at Monash University, IITB-Monash University Academy funding, and the Department of Biotechnology within the Government of India (#BT/PR13442/MED/32/440/2015).

# Additional information

## Funding

| Funder | Grant reference number | Author |
|---|---|---|
| Australian Research Council | DP190100343 | Reza Nosrati<br>Ranganathan Prabhakar |
| Australian Research Council | DP200100659 | Moira K O'Bryan |
| Department of Biotechnology, Ministry of Science and Technology | BT/PR13442/MED/32/440/2015 | Sameer Jadhav |

The funders had no role in study design, data collection and interpretation, or the decision to submit the work for publication.

## Author contributions

Ashwin Nandagiri, Data curation, Software, Formal analysis, Validation, Investigation, Visualization, Methodology, Writing - original draft, Writing - review and editing; Avinash Satish Gaikwad, Data curation, Formal analysis, Validation, Investigation, Methodology, Writing - review and editing; David L Potter, Resources, Data curation, Software, Methodology, Project administration, Writing - review and editing; Reza Nosrati, Resources, Methodology, Project administration, Writing - review and editing; Julio Soria, Resources, Formal analysis, Methodology, Project administration, Writing - review and editing; Moira K O'Bryan, Conceptualization, Resources, Formal analysis, Supervision, Funding acquisition, Investigation, Methodology, Project administration, Writing - review and editing; Sameer Jadhav, Conceptualization, Resources, Formal analysis, Supervision, Funding acquisition, Methodology, Project administration, Writing - review and editing; Ranganathan Prabhakar, Conceptualization, Resources, Formal analysis, Supervision, Funding acquisition, Methodology, Writing - original draft, Project administration, Writing - review and editing

## Author ORCIDs

Ashwin Nandagiri (iD) https://orcid.org/0000-0001-7328-9288
Avinash Satish Gaikwad (iD) https://orcid.org/0000-0002-7379-6383
Reza Nosrati (iD) https://orcid.org/0000-0002-1461-229X
Julio Soria (iD) https://orcid.org/0000-0002-7089-9686
Moira K O'Bryan (iD) http://orcid.org/0000-0001-7298-4940
Sameer Jadhav (iD) https://orcid.org/0000-0002-4207-3393
Ranganathan Prabhakar (iD) https://orcid.org/0000-0001-7357-4222

## Ethics

Animal experimentation: This study was performed in strict accordance with the Australian Code of Practice for the Care and Use of Animals for Scientific Purposes. All of the animals were handled according to institutional animal care and use protocols approved by the Monash Animal Ethics committee (Approval # MARP/2014/084).

## Decision letter and Author response

Decision letter https://doi.org/10.7554/eLife.62524.sa1
Author response https://doi.org/10.7554/eLife.62524.sa2

# Additional files

## Supplementary files

- Transparent reporting form

## Data availability

Source data files for all results figures have been provided. Videos of the WT and KO mice sperm samples are available for public access and download from the Monash University Research Repository (DOI: https://doi.org/10.26180/5f50562bb322b) MATLAB Codes used to analyze the data to produce the results in the manuscript are available for public access and download from the Monash University Research Repository (DOI: https://doi.org/10.26180/14045816).

The following datasets were generated:

| Author(s) | Year | Dataset title | Dataset URL | Database and Identifier |
|---|---|---|---|---|
| Nandagiri A | 2021 | MATLAB scripts for Chebyshev-POD/flagellar energetics calculations | https://doi.org/10.26180/14045816 | Monash University Research Repository, 10.26180/14045816 |
| Nandagiri A, Gaikwad AS, Potter DL, Nosrati R, Soria J, O'Bryan MK, Jadhav S, Prabhakar R | 2020 | Flagellar energetics from high-resolution imaging of beating patterns in tethered mouse sperm | https://doi.org/10.26180/5f50562bb322b | Monash University Research Repository, 10.26180/5f50 562bb322b |

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

## Appendix 1

### Conservation laws

The position vector of any material point on a cross-section is $\mathbf{x} = \mathbf{r} + \mathbf{R}$, where $\mathbf{r}$ is the point on the cross-section through which the filament axis passes and $\mathbf{R}$ is the vector displacement of the material point from the axial point. Then, the velocity of the material point is $\dot{\mathbf{x}} = \dot{\mathbf{r}} + \dot{\mathbf{R}}$. Cross-sectional planes can rotate relative to each other. Then, $\partial \mathbf{d}_i / \partial t = \omega \times \mathbf{d}_i$, where $\omega(s,t)$ is the instantaneous angular velocity of the cross-sectional plane through the axial point at $s$. The velocity of any material point, $\partial \mathbf{x} / \partial t = \mathbf{v} + \partial \mathbf{R} / \partial t$, where $\mathbf{v} = \partial \mathbf{r} / \partial t$. With this, we have $\partial \mathbf{R} / \partial t = \omega \times \mathbf{R}$. It can further be shown that

$$\frac{\partial \omega}{\partial s} = \frac{\partial \Omega_i}{\partial t} \mathbf{d}_i, \tag{48}$$

where Einstein's summation convention is used. This implies that, for planar motion, where $\omega = \omega \mathbf{e}_z$,

$$\frac{\partial \omega}{\partial s} = \frac{\partial C}{\partial t}. \tag{49}$$

Using kinematic relationships and the Frenet–Serret equations, it can be shown that

$$\frac{\partial \omega}{\partial s} = \frac{\partial C}{\partial t} \mathbf{b} + \frac{\partial T}{\partial t} \mathbf{t}. \tag{50}$$

Mass conservation is trivially satisfied for an inextensible rod whose density is constant and whose cross-sectional area is independent of time. The net hydrodynamic force on a cross-section,

$$\mathbf{f}^{\mathrm{h}}(s,t) = \int_{\Gamma_e(s)} \tau^{\mathrm{h}} \, d\ell_e. \tag{51}$$

Here, $\tau^{\mathrm{h}}$ is the hydrodynamic traction acting on the external surface and $\Gamma_e(s)$ is the external perimeter of the cross-section at any $s$, parameterized by an arc-length variable along the perimeter, $\ell_e$. The external force distribution $\mathbf{f}^e(s,t)$ similarly accounts for non-hydrodynamic surface traction such as that due to wall contact. The axonemal motors exert forces on the *internal* surfaces of the passive flagellar material. The surface traction, $\tau^{\mathrm{a}}$, exerted by these motors results in an active force distribution,

$$\mathbf{f}^{\mathrm{a}}(s,t) = \int_{\Gamma_i(s)} \tau^{\mathrm{a}} \, d\ell_i, \tag{52}$$

where $\Gamma_i(s)$ is the perimeter at any section $s$ of the internal surfaces and $\ell_i$ is an arc-length variable along that perimeter. The passive material stress tensor $\sigma$ acting throughout the cross-sectional domain $\Sigma(s)$ results in a net force, $\mathbf{F}$, on a cross-section by the material on its aft side:

$$\mathbf{F}(s,t) = \int_{\Sigma(s)} \sigma \cdot \mathbf{t} \, d\mathcal{A}, \tag{53}$$

where $d\mathcal{A}$ is a differential area element on a cross-section at $s$, and for an unshearable rod, the aft-side outward unit normal $\mathbf{d}_1$ at any cross-section is identical to the centerline unit tangent vector, $\mathbf{t} = \partial \mathbf{r} / \partial s$. The torques due to the hydrodynamic and motor tractions are

$$\mathbf{m}^{\mathrm{h}}(s,t) = \int_{\Gamma_e(s)} \mathbf{R}_e \times \tau^{\mathrm{h}} \, d\ell_e; \quad \mathbf{m}^{\mathrm{a}} = \int_{\Gamma_i(s)} \mathbf{R}_i \times \tau^{\mathrm{a}} \, d\ell_i. \tag{54}$$

The torque distribution, $\mathbf{m}^e$, due to the external non-hydrodynamic surface traction is similarly defined. The torque due exerted by the material on the aft side,

$$\mathbf{M}(s,t) = \int_{\Sigma(s)} \mathbf{R} \times \sigma \cdot \mathbf{t} \, d\mathcal{A}. \tag{55}$$

The conservation of linear momentum for a section of the rod from $s_1$ to $s_2$ is given by

$$\frac{d}{dt}\int_{s_1}^{s_2}\int_{\Sigma(s)}\rho\dot{x}d\mathcal{A}\,ds = \int_{s_1}^{s_2}\int_{\Gamma_e(s)}\tau^h d\ell_e ds + \int_{s_1}^{s_2}\int_{\Gamma_i(s)}\tau^a d\ell_i ds + \int_{s_1}^{s_2}\int_{\Gamma_e(s)}\tau^e d\ell_e ds$$

$$-\int_{\Sigma(s_1)}\sigma\cdot t d\mathcal{A} + \int_{\Sigma(s_2)}\sigma\cdot t d\mathcal{A},$$

$$= \int_{s_1}^{s_2} f^h\,ds + \int_{s_1}^{s_2} f^a\,ds + \int_{s_1}^{s_2} f^e\,ds - F(s_1) + F(s_2).$$

For an inertialess rod, the differential form of the equation above is

$$\mathbf{f}^h + \mathbf{f}^a + \mathbf{f}^e + \frac{\partial \mathbf{F}}{\partial s} = \mathbf{0}. \tag{56}$$

The gradient with respect to $s$ of $\mathbf{F}$ in the momentum balance thus describes the net force per unit length at a cross-section due to the passive internal stress.

Similarly, the conservation of angular momentum for a section of the rod can be written as follows:

$$\frac{d}{dt}\int_{s_1}^{s_2}\int_{\Sigma(s)}x\times\rho\underline{x}d\mathcal{A}\,ds = \int_{s_1}^{s_2}\int_{\Gamma_e(s)}(r+R_e)\times\tau^h d\ell_e\,ds + \int_{s_1}^{s_2}\int_{\Gamma_i(s)}(r+R_i)\times\tau^a d\ell_i\,ds$$

$$+ \int_{s_1}^{s_2}\int_{\Gamma_i(s)}(r+R_e)\times\tau^e d\ell_i\,ds$$

$$-\int_{\Sigma(s_1)}(r+R)\times\sigma\cdot t d\mathbf{A} + \int_{\Sigma(s_2)}(r+R)\times\sigma\cdot t d\mathcal{A},$$

$$= \int_{s_1}^{s_2} r\times f^h\,ds + \int_{s_1}^{s_2} r\times f^a\,ds$$

$$+ \int_{s_1}^{s_2} r\times f^e\,ds - r(s_1)\times F(s_1) + r(s_2)\times F(s_2) - M(s_1) + M(s_2),$$

which leads to the following differential equation for an inertialess rod after eliminating terms using the differential form of the linear momentum equation earlier:

$$\mathbf{m}^h + \mathbf{m}^a + \mathbf{m}^e + \mathbf{t}\times\mathbf{F} + \frac{\partial \mathbf{M}}{\partial s} = \mathbf{0}. \tag{57}$$

The First Law of Thermodynamics provides an equation that balances the rate of energy change with the work done and heat input to a control volume. In the case of an internally driven rod, the total energy is the sum of the passive elastic energy, the thermal internal energy, and the kinetic energy. Work is done on the control volume by the surface tractions exerted by the surrounding fluid, the internal motors, and the external tethering constraint. Work is also done by the passive material stress on the cross-sections. Heat can be transferred out of the control volume to the surroundings. The conservation of energy implies

$$\frac{d}{dt}\int_{s_1}^{s_2}\int_{\Sigma(s)}(\frac{1}{2}\rho\dot{x}^2 + \hat{u} + \hat{\epsilon})\,d\mathcal{A}\,ds =$$

$$\int_{s_1}^{s_2}\int_{\Gamma_e(s)}(v+\omega\times R_e)\cdot(\tau^h+\tau^e)\,d\ell_e\,ds + \int_{s_1}^{s_2}\int_{\Gamma_e(s)}(v+\omega\times R_i)\cdot\tau^a d\ell_i\,ds \tag{58}$$

$$-\int_{\Sigma(s_1)}(v+\omega\times R)\cdot\sigma\cdot t d\mathcal{A} + \int_{\Sigma(s_2)}(v+\omega\times R)\cdot\sigma\cdot t_2\,d\mathcal{A} - \int_{s_1}^{s_2} q\,ds.$$

Here, $\hat{u}$ and $\hat{\epsilon}$ are the thermal internal energy and elastic strain energy densities, and $q$ is the heat transferred per unit length out of the cross-section at $s$. Defining the energy distributions $\epsilon = \int_{\Sigma(s)}\hat{\epsilon}d\mathcal{A}$ and $u = \int_{\Sigma(s)}\hat{u}d\mathcal{A}$, and neglecting kinetic energy changes in an inertialess rod, the differential form of the energy equation is obtained as

$$\frac{\partial \epsilon}{\partial t} + \frac{\partial u}{\partial t} = \mathbf{v}\cdot(\mathbf{f}^h+\mathbf{f}^e) + \mathbf{v}\cdot\mathbf{f}^a + \omega\cdot(\mathbf{m}^h+\mathbf{m}^e) + \omega\cdot\mathbf{m}^a + \frac{\partial(\mathbf{v}\cdot\mathbf{F})}{\partial s} + \frac{\partial(\omega\cdot\mathbf{M})}{\partial s} - q. \tag{59}$$

Defining the power densities due to the hydrodynamic, external, and active forces and moments as

$$p^{\mathrm{hd}} = \mathbf{v} \cdot \mathbf{f}^{\mathrm{h}} + \omega \cdot \mathbf{m}^{\mathrm{h}}; \quad p^{\mathrm{e}} = \mathbf{v} \cdot \mathbf{f}^{\mathrm{e}} + \omega \cdot \mathbf{m}^{\mathrm{e}}; \quad p^{\mathrm{a}} = \mathbf{v} \cdot \mathbf{f}^{\mathrm{a}} + \omega \cdot \mathbf{m}^{\mathrm{a}}, \tag{60}$$

and the net rate of work done on a cross-section by the action of the local stress gradient as

$$p^{\mathrm{s}} = \frac{\partial (\mathbf{v} \cdot \mathbf{F})}{\partial s} + \frac{\partial (\omega \cdot \mathbf{M})}{\partial s}, \tag{61}$$

we obtain,

$$\frac{\partial \epsilon}{\partial t} + \frac{\partial u}{\partial t} = p^{\mathrm{a}} + p^{\mathrm{hd}} + p^{\mathrm{e}} + p^{\mathrm{s}} - q. \tag{62}$$

*Equations (56), (57), and (62)* are the point-wise linear and angular momentum, and energy, balances for an internally driven Kirchhoff rod.

## Passive, rigid, tethered head

Since the head is passive, $\mathbf{f}^{\mathrm{a}} = \mathbf{m}^{\mathrm{a}} = \mathbf{0}$. A rigid body cannot deform and store energy elastically. There is also no heat transfer between the body and its surroundings. Hence, the equations governing the dynamics of the head region are

$$\mathbf{f}^{\mathrm{h}} + \mathbf{f}^{\mathrm{e}} + \frac{\partial \mathbf{F}}{\partial s} = \mathbf{0}; \tag{63}$$

$$\mathbf{m}^{\mathrm{h}} + \mathbf{m}^{\mathrm{e}} + \mathbf{t} \times \mathbf{F} + \frac{\partial \mathbf{M}}{\partial s} = \mathbf{0}; \tag{64}$$

$$0 = p^{\mathrm{hd}} + p^{\mathrm{e}} + p^{\mathrm{s}}. \tag{65}$$

The linear and angular momentum equations above can, in principle, be integrated for $\mathbf{F}$ and $\mathbf{M}$ if the hydrodynamic and external forces and moments are known. The boundary conditions at the free end at $s = 0$ are $\mathbf{F}(0,t) = \mathbf{M}(0,t) = \mathbf{0}$. At the neck, $s = s_{\mathrm{n}}$, $\mathbf{F}(s_{\mathrm{n}},t) = \mathbf{F}_{\mathrm{n}}$, $\mathbf{M}(s_{\mathrm{n}},t) = \mathbf{M}_{\mathrm{n}}$, where $\mathbf{F}_{\mathrm{n}}$ and $\mathbf{M}_{\mathrm{n}}$ are the internal force and bending moment due to the passive stress at the neck. These are calculated from the tail side since material stress is continuous across the neck junction.

In this work, however, we only use the energy equation. Integrating it across the head region, using *Equation (61)* and the boundary conditions, and rearranging, we obtain

$$P_{\mathrm{h}}^{\mathrm{hd}} + P_{\mathrm{h}}^{\mathrm{e}} = (\mathbf{v}_0 \cdot \mathbf{F}_0 + \omega_0 M_0) - (\mathbf{v}_{\mathrm{N}} \cdot \mathbf{F}_{\mathrm{N}} + \omega_{\mathrm{N}} M_{\mathrm{N}}) = -(\mathbf{v}_{\mathrm{N}} \cdot \mathbf{F}_{\mathrm{N}} + \omega_{\mathrm{N}} M_{\mathrm{N}}). \tag{66}$$

Here, $P_{\mathrm{h}}^{\mathrm{hd}}$ and $P_{\mathrm{h}}^{\mathrm{e}}$ are the total instantaneous power exerted on the head by the hydrodynamic and tethering forces, respectively, and $\mathbf{v}_{\mathrm{N}}$ and $\omega_{\mathrm{N}}$ are the velocities at the neck.

## Active, viscoelastic, untethered tail

No external forces act on the freely beating tail. No net active force is exerted by the dynein motors at any cross-section. The contribution of the hydrodynamic moment is negligible, Hence, $\mathbf{f}^{\mathrm{e}} = \mathbf{f}^{\mathrm{a}} = \mathbf{m}^{\mathrm{h}} = \mathbf{m}^{\mathrm{e}} = \mathbf{0}$. As noted in the main text, the passive internal stress in the tail can be split into contributions from bending elasticity and bending friction. With his split, it is shown in the main text that, for the system to remain isothermal, the heat generated by internal friction, $p^{\mathrm{id}}$, must be balanced by the heat transfer rate, $q$. With all these substitutions, the conservation equations for the tail region are thus

$$\mathbf{f}^{\mathrm{h}} + \frac{\partial \mathbf{F}}{\partial s} = \mathbf{0}; \tag{67}$$

$$\mathbf{m}^{\mathrm{a}} + \mathbf{t} \times \mathbf{F} + \frac{\partial \mathbf{M}}{\partial s} = \mathbf{0}; \tag{68}$$

$$\frac{\partial \epsilon}{\partial t} = p^{\mathrm{a}} + p^{\mathrm{hd}} + p^{\mathrm{s}} + p^{\mathrm{id}}. \tag{69}$$

In this work, we are interested in the inverse problem of determining $p^{\mathrm{a}}$ from the observed kinematics. As noted in the main text, the passive internal force in the tail is formally calculated from the hydrodynamic force distribution as

$$\mathbf{F}(s,t) = \int_s^L \mathbf{f}^{\mathrm{h}}(s',t)\,ds'. \tag{70}$$

This satisfies the condition that $\mathbf{F}(L,t) = 0$. However, since we do not have motion data for $s > s_{\mathrm{t}}$, we neglect the contribution to the hydrodynamic forces from $s > s_{\mathrm{t}}$. This effectively amounts to applying the force-free condition at $s = s_{\mathrm{t}}$. The force- and torque-free conditions at the tail end are applied to calculate the instantaneous power balance across the entire tail region. Integrating the energy balance over the tail region, we obtain

$$\dot{E} = P^{\mathrm{a}} + P^{\mathrm{hd}} + P^{\mathrm{id}} + (\mathbf{v}_L \cdot \mathbf{F}_L + \omega_L M_L) - (\mathbf{v}_{\mathrm{N}} \cdot \mathbf{F}_{\mathrm{N}} + \omega_{\mathrm{N}} M_{\mathrm{N}}). \tag{71}$$

The free-end boundary condition is used to set $\mathbf{F}_L = \mathbf{0}$ and $M_L = 0$. Using *Equation (66)* earlier, the neck contribution can be replaced in terms of the total power dissipation by the head, $P_{\mathrm{h}}^{\mathrm{d}} = P_{\mathrm{h}}^{\mathrm{hd}} + P_{\mathrm{h}}^{\mathrm{e}}$. Further, we split the integral of $p^{\mathrm{a}}$ into contributions from its negative and positive parts by defining the motor dissipation and motor input as follows:

$$P^{\mathrm{md}}(t) = \int_{s_{\mathrm{n}}}^L \min(p^{\mathrm{a}}, 0)\,ds; \quad P^{\mathrm{mi}}(t) = \int_{s_{\mathrm{n}}}^L \max(p^{\mathrm{a}}, 0)\,ds. \tag{72}$$

Substituting these in *Equation (71)* and rearranging, we obtain (noting that dissipations are negative in our sign convention)

$$P^{\mathrm{mi}} = \dot{E} - P^{\mathrm{hd}} - P^{\mathrm{id}} - P^{\mathrm{md}} - P_{\mathrm{h}}^{\mathrm{d}}. \tag{73}$$

In other words, the motor input provided is either used for elastic storage or to overcome all the contributions to dissipation.

The equations above show that the energy balance equations used in the calculations described in the main text are formally consistent with the free-end conditions. It also shows that we have effectively neglected contributions from the non-imaged tail end. We do not, however, expect significant qualitative changes due to this approximation.

## Appendix 2

### Planar beating of tethered mouse sperm

A detailed study by *Woolley, 2003* showed that freely swimming mouse sperm exhibit preferential capture by walls. They are hydrodynamically drawn to walls and then stabilize with *the left sides of their flat heads held against the surface*. If a sperm arrives at a wall with its right side next to the surface, it quickly moves away from the wall after a few flagellar beats and will be eventually captured in its stable orientation. Sperm following this 'left-side rule' exhibit planar beating in a plane that is parallel to the wall. This beating is clearly resolvable within the focal width of the microscope. On the other hand, those that approach the wall with the right sides of their heads parallel to the wall exhibit considerable motion out of the focal plane. Scanning electron microscopy further revealed that the plane of the left side of the flat head makes an angle less than 180° with the flagellum at the neck. This appears to enable sperm following the left-side rule to stabilize planar beating parallel to the wall.

In our samples, the heads are chemically tethered to the wall. While we observe a few cells adhere on their right sides, these cells exhibit non-planar beating out of the focal plane. *Appendix 2—figure 1* sketches the expected geometry of the mouse sperm cell with its intrinsic head-tail angle at the neck, when the left side of its head is tethered to the wall. In this orientation, the flagellum beats in a plane parallel to the wall and its centerline is at a distance equal to the neck radius, $h = a_\mathrm{n}$, from the wall, where $a_\mathrm{n}$ is the radius of the neck. The head-neck angle is the same as the angle made by the head axis with the wall, $\theta \approx a_\mathrm{n}/\ell_\mathrm{h}$, where $\ell_\mathrm{h}$ is the length of the head. From literature measurements of mouse sperm dimensions, $a_\mathrm{n} \approx 0.6\ \mu\mathrm{m}$ and $\ell_\mathrm{h} \approx 5\ \mu\mathrm{m}$, which gives $\theta \approx 0.1$ rad or 6°. The neck radius, $a_\mathrm{n}$, is 0.57 $\mu\mathrm{m}$, and the radius, $a_\mathrm{t}$, of the filament at the tail end is 0.18 $\mu\mathrm{m}$ (*Gu et al., 2019*).

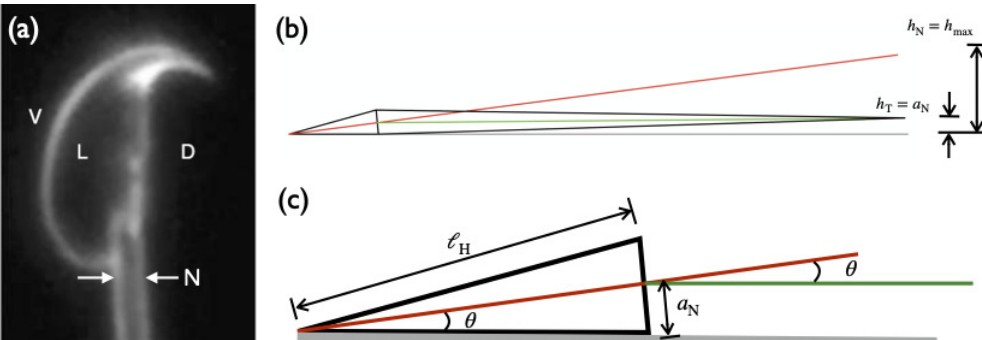

**Appendix 2—figure 1.** Orientation of the sperm cell with respect to the glass slide. (**a**) Image from *Woolley, 2003* showing the left side (L) of a mouse sperm head facing the viewer with the ventral (V) and dorsal (D) sides of the head indicated. The concave side of the hook is towards the dorsal side. Also shown is the neck (N) at the proximal end of the flagellum. (**b**) Schematic showing the mouse sperm body as viewed from its dorsal side when the left side of its head is against the wall. The intrinsic angle made by the head with the flagellum at the neck enables planar beating when the left side of the head is against the wall. The red and green lines indicate the axes of the head and the tail. In this orientation, the flagellum beats in a plane parallel to the wall and its centerline is at a distance equal to the neck radius $,a_\mathrm{n}$, from the wall. If there had been no angle at the neck, the tip of the tail would be at a height of $h_\mathrm{t} = h_\mathrm{max}$. (**c**) The angle (in radians) of the head axis to the wall, $\theta \approx a_\mathrm{n}/\ell_\mathrm{h}$.

We have also taken advantage of the nearly planar beating that many mammalian sperm exhibit near walls. In our experiments, out-of-plane excursions in the resolved portion of the tail appear limited to less than 2 µm. Nearly planar beating of the mid-piece and principal piece is evidenced by the fact that these portions of the flagellum remained in focus at all times in our samples. The total depth of field

$$d_\mathrm{tot} = \frac{\lambda\, n}{\mathrm{NA}^2} + \frac{n\, e}{M\, \mathrm{NA}},$$

(74)

is 1.2 μm in our microscope, where $\lambda = 0.7\ \mu$m is the mean wavelength of the incident light, $n = 1$ is the refractive index of the air medium between the coverslip and the objective lens, $M = 20$ is the lateral magnification, $\mathrm{NA} = 0.7$ is the numerical aperture of the objective, and $e = 0.65\ \mu$m is the resolution of the detector in the image plane. After taking into account uncertainties due to the spread in incident wavelengths and other factors, we estimate that the depth of field cannot be larger than 2 $\mu$m. We have verified this independently in calibration experiments using 5 $\mu$m diameter spherical particles adhered to the bottom surface.

A maximum vertical deviation of around 2 μm is about 10% of the mean amplitude of the in-plane beating of around 20 μm. Therefore, neglecting such deviations can be expected to contribute errors of around 10% in the velocity components, curvature, and rate of curvature. These errors propagate quadratically when calculating energetic quantities. The resulting error due to non-planarity in beating cannot therefore be larger than 1%, which is considerably smaller than the natural fluctuations in the beating patterns within a single sample and the sample-to-sample variations.

# Appendix 3

**Appendix 3—table 1.** One-way ANOVA data for establishing that sample means of cycle variables are significantly different from population means in the wildtype (WT) and knockout (KO) genotypes; $F$ denotes the $F$-test statistic and $p$ denotes probability of the null hypothesis.

| Genotype | Flagellar region | Power | Treatments DOF | Error DOF | $F$ | $p$ |
|---|---|---|---|---|---|---|
| WT | Full | Cycle time | 4 | 306 | 50.31 | $<10^{-16}$ |
| WT | Full | Input power | 4 | 306 | 833.19 | $<10^{-16}$ |
| WT | Full | Hyd. dissn. | 4 | 306 | 1441.37 | $<10^{-16}$ |
| WT | Full | Internal dissn. | 4 | 306 | 730.68 | $<10^{-16}$ |
| WT | Full tail | Motor dissn. | 4 | 306 | 246.17 | $<10^{-16}$ |
| WT | Mid-piece | Input power | 4 | 306 | 483.02 | $<10^{-16}$ |
| WT | Mid-piece | Hyd. dissn. | 4 | 306 | 186.86 | $<10^{-16}$ |
| WT | Mid-piece | Internal dissn. | 4 | 306 | 436.92 | $<10^{-16}$ |
| WT | Mid-piece | Motor dissn. | 4 | 306 | 421.04 | $<10^{-16}$ |
| WT | Principal piece | Input power | 4 | 306 | 786.93 | $<10^{-16}$ |
| WT | Principal piece | Hyd. disspn. | 4 | 306 | 1716.31 | $<10^{-16}$ |
| WT | Principal piece | Internal dissn. | 4 | 306 | 520.19 | $<10^{-16}$ |
| WT | Principal piece | Motor dissn. | 4 | 306 | 140.96 | $<10^{-16}$ |
| KO | Full tail | Cycle time | 4 | 270 | 235.45 | $<10^{-16}$ |
| KO | Full tail | Input power | 4 | 270 | 110.04 | $<10^{-16}$ |
| KO | Full tail | Hyd. dissn. | 4 | 270 | 198.37 | $<10^{-16}$ |
| KO | Full tail | Internal dissn. | 4 | 270 | 90.02 | $<10^{-16}$ |
| KO | Full tail | Motor dissn. | 4 | 270 | 120.31 | $<10^{-16}$ |
| KO | Mid-piece | Input power | 4 | 270 | 528.12 | $<10^{-16}$ |
| KO | Mid-piece | Hyd. dissn. | 4 | 270 | 216.51 | $<10^{-16}$ |
| KO | Mid-piece | Internal dissn. | 4 | 270 | 247.53 | $<10^{-16}$ |
| KO | Mid-piece | Motor dissn. | 4 | 270 | 374.02 | $<10^{-16}$ |
| KO | Principal piece | Input power | 4 | 270 | 125.73 | $<10^{-16}$ |
| KO | Principal piece | Hyd. dissn. | 4 | 270 | 206.07 | $<10^{-16}$ |
| KO | Principal piece | Internal dissn. | 4 | 270 | 109.09 | $<10^{-16}$ |
| KO | Principal piece | Motor dissn. | 4 | 270 | 20.3 | $<10^{-16}$ |

**Appendix 3—table 2.** Comparison of the external hydrodynamic dissipation with the internal frictional and motor dissipations: $t$ denotes the Student's $t$-test statistic in an unpaired, two-tailed test; $p$ is the probability of the null hypothesis.

| Genotype | Flagellar region | $\eta_n$ (Pa s $\mu$m$^4$) | Hyd. dissn. (fW) | Int. dissn. (fW) | $t$ | $p$ |
|---|---|---|---|---|---|---|
| WT | Full tail | $10^3$ | 89.07 | 240.05 | 22.71 | $<10^{-16}$ |
| WT | Mid-piece | $10^3$ | 6.69 | 108.43 | 36.39 | $<10^{-16}$ |
| WT | Principal piece | $10^3$ | 82.21 | 131.11 | 9.82 | $<10^{-16}$ |
| KO | Full tail | $10^3$ | 66.39 | 200.91 | 21.44 | $<10^{-16}$ |
| KO | Mid-piece | $10^3$ | 4.80 | 32.40 | 29.16 | $<10^{-16}$ |
| KO | Principal piece | $10^3$ | 61.48 | 168.28 | 16.73 | $<10^{-16}$ |

*Continued on next page*

| Genotype | Flagellar region | $\eta_n$ (Pa s $\mu$m$^4$) | Hyd. dissn. (fW) | Motor dissn. (fW) | $t$ | $p$ |
|---|---|---|---|---|---|---|
| WT | Full tail | $10^3$ | 89.07 | 135.06 | 11.48 | $<10^{-16}$ |
| WT | Mid-piece | $10^3$ | 6.69 | 109.75 | 36.21 | $<10^{-16}$ |
| WT | Principal piece | $10^3$ | 82.21 | 25.09 | 20.71 | $<10^{-16}$ |
| KO | Full tail | $10^3$ | 66.39 | 48.72 | 9.74 | $<10^{-16}$ |
| KO | Mid-piece | $10^3$ | 4.80 | 29.14 | 24.87 | $<10^{-16}$ |
| KO | Principal piece | $10^3$ | 61.48 | 19.49 | 28.4 | $<10^{-16}$ |
| WT | Full tail | 0 | 89.07 | 155.58 | 11.48 | $<10^{-16}$ |
| WT | Mid-piece | 0 | 6.69 | 112.56 | 36.21 | $<10^{-16}$ |
| WT | Principal piece | 0 | 82.21 | 42.71 | 20.71 | $<10^{-16}$ |
| KO | Full tail | 0 | 66.39 | 77.61 | 9.74 | $<10^{-16}$ |
| KO | Full tail | 0 | 4.80 | 35.23 | 24.87 | $<10^{-16}$ |
| KO | Principal piece | 0 | 61.48 | 42.28 | 28.4 | $<10^{-16}$ |

**Appendix 3—table 3.** Comparison of wildtype (WT) and knockout (KO) samples.
The values in in columns 2 and 3 are the means of the distributions created by pooling together values from the individual cycles of all the sperm samples in each genotype.

| Quantity | WT | KO | $t$ | $p$ |
|---|---|---|---|---|
| Cycle time (s) | 0.16 | 0.18 | 2.72 | 0.01 |
| Reciprocal cycle time (Hz) | 7.19 | 7.2 | 0.03501 | 0.97 |
| Full tail hyd. dissn. (fW) | 89.07 | 66.39 | 6.91 | $1.26 \times 10^{-11}$ |
| Mid-piece hyd. dissn. (fW) | 6.69 | 4.80 | 6.89 | $1.49 \times 10^{-11}$ |
| Principal piece hyd. dissn. (fW) | 82.21 | 61.48 | 6.71 | $4.63 \times 10^{-11}$ |
| Full tail motor dissn. (fW), $\eta_n = 10^3$ Pa s $\mu$m$^4$ | 135.06 | 48.72 | 26.89 | $<10^{-16}$ |
| Full tail int. dissn. (fW), $\eta_n = 10^3$ Pa s $\mu$m$^4$ | 240.05 | 200.91 | 4.55 | $6.54 \times 10^{-6}$ |
| Mid-piece motor disspn. (fW), $\eta_n = 10^3$ Pa s $\mu$m$^4$ | 109.75 | 29.14 | 25.57 | $<10^{-16}$ |
| Mid-piece int. dissn. (fW), $\eta_n = 10^3$ Pa s $\mu$m$^4$ | 108.43 | 32.40 | 24.59 | $<10^{-16}$ |
| Principal piece motor dissn. (fW), $\eta_n = 10^3$ Pa s $\mu$m$^4$ | 25.08 | 19.49 | 5.62 | $3.01 \times 10^{-8}$ |
| Principal piece int. dissn. (fW), $\eta_n = 10^3$ Pa s $\mu$m$^4$ | 131.11 | 168.28 | 5.03 | $3.01 \times 10^{-8}$ |
| Full tail motor dissn. (fW), $\eta_n = 0$ Pa s $\mu$m$^4$ | 155.58 | 77.61 | 24.87 | $<10^{-16}$ |
| Mid-piece motor dissn. (fW), $\eta_n = 0$ Pa s $\mu$m$^4$ | 112.56 | 35.23 | 29.10 | $<10^{-16}$ |
| Principal piece motor dissn. (fW), $\eta_n = 0$ Pa s $\mu$m$^4$ | 42.71 | 42.28 | 0.22 | 0.82 |

