## [Decision Letter]

**Acceptance summary:**

This study combines high resolution imaging experiments with mechanical modeling to elucidate the energetics of flagellar propulsion and understand the role of internal dissipation in this system. The authors conclude that the main origin of dissipation is internal to the flagella, a finding that challenges the conventional view of flagellar dynamics, and should prove to be of interest to a wide range of researchers.

**Decision letter after peer review:**

Thank you for submitting your article "Flagellar energetics from high-resolution imaging of beating patterns in tethered mouse sperm" for consideration by *eLife*. Your article has been reviewed by 2 peer reviewers, one of whom is a member of our Board of Reviewing Editors, and the evaluation has been overseen by Aleksandra Walczak as the Senior Editor. The reviewers have opted to remain anonymous.

The reviewers have discussed the reviews with one another and the Reviewing Editor has drafted this decision to help you prepare a revised submission.

As the editors have judged that your manuscript is of interest, but as described below that additional analysis is required before it is published, we would like to draw your attention to changes in our revision policy that we have made in response to COVID-19 (https://elifesciences.org/articles/57162). First, because many researchers have temporarily lost access to the labs, we will give authors as much time as they need to submit revised manuscripts. We are also offering, if you choose, to post the manuscript to bioRxiv (if it is not already there) along with this decision letter and a formal designation that the manuscript is "in revision at *eLife*". Please let us know if you would like to pursue this option. (If your work is more suitable for medRxiv, you will need to post the preprint yourself, as the mechanisms for us to do so are still in development.)

Summary:

This study combines high resolution imaging experiments with mechanical modeling to elucidate the energetics of flagellar propulsion and understand the role of internal dissipation in this system. The experiments use mouse sperm cells that are chemically tethered to a glass slip. For each cell, the flagellum shape is imaged over time and segmented into a mathematical curve. This data is analyzed based on a planar Kirckhoff rod model that includes hydrodynamic drag forces (based on resistive force theory), bending elasticity, and an unknown active moment density. An energy balance is written that also includes internal viscous dissipation generated inside the flagellum, with an ad hoc internal friction coefficient. By calculating the various terms in the energy balance based on the reconstructed filament shapes, the authors are able to estimate the active power density along the flagellum. This calculation leads to two unexpected findings: (1) the authors find that the active power density can be negative along some portions of the flagellum, meaning that along these portions the dynein motors act against the local deformation of the structure, and (2) the main origin of dissipation in the system comes from internal dissipation, which exceeds viscous dissipation in the fluid in magnitude.

Essential revisions:

1. It is not completely clear from the manuscript what the configuration of the sperm is with respect to the glass slide where the head is tethered. What is the orientation of the cells with respect to the slide, and in which plane are the deformations measured? (from above or from the side?) We would expect that different configurations may lead to slightly different waveforms. In particular, we are surprised that the mean shapes shown in figure 2(a) have a net asymmetry which is observed in nearly all the cells: could this have to do with the relative configuration of the flagellum with respect to the surface?

2. The experiments are done with flagella very near a no-slip surface, since the cells are chemically adhered to the chamber boundary. Yet, the authors use resistive force theory for filaments in free space, without any reference to the nearby no-slip surface. As the rate of energy dissipation near the surface will be considerably larger than estimated by RFT, it is possible that some (or much, or perhaps all) of the additional dissipation found by the authors is actually within the fluid and simply not accounted for by RFT. Thus, all of the calculations must be redone with the appropriate Blake tensor for stokeslets near a no-slip wall before the results can be considered definitive. The paper must also more carefully illustrate and quantify the proximity of the flagella to the surface in order to make these calculations precise. Absent this analysis, the claims of the paper do not stand up to scrutiny.

A related point is the need to understand the effect of tethering the cell on its kinematics and energetics? In other words, do the conclusions still hold for freely swimming cells?

3. Is there any evidence of 3D dynamics? Some recent experiments with human sperm have suggested that sperm beats can take place in 3D (Gadelha et al., Science Advances 2020). As the model in the paper is 2D, this could also affect the energy balance.

4. The authors should examine the work of K.E. Machin ["The control and synchronization of flagellar movement"], Proc. Roy. Soc. B 158, 88 (1963), which provided the first theoretical formalism to study active moment generation within beating flagella based on examining the difference between known force contributions from viscous dissipation and elastic bending. It seems that this same kind of analysis could be done here to identify directly the non-viscous contribution, rather than having to postulate a particular form.

Stated another way: Why not try to estimate the active power density directly from the active moment density, which could be calculated from the moment balance of equation (4) where all the other terms are known? This would provide a direct estimate of the active power. The force balance could then be used to estimate the internal friction, which would then no longer rely on an assumed value for the internal friction coefficient. In fact, this could be used to obtain an estimate for that coefficient.

5. The paper addresses in detail the use of Chebyshev fitting methods for the filaments, but does not appear to address the physical boundary conditions one would expect on elastic objects (particularly at the free end), involving the vanishing of moments and forces. Unlike, for example, the biharmonic eigenfunctions of simple elastic filament dynamics which are tailored to those boundary conditions [see, e.g. Goldstein, Powers, Wiggins, PRL 80, 5232 (1998)], it is not clear how the Chebyshev functions satisfy those conditions. Some explanation is needed.

6. If indeed internal dissipation dominates, that would suggest that essentially all prior theoretical approaches to calculating sperm waveforms must be quantitatively in error by very large factors. It would be very appropriate for the authors to examine some of those theoretical works to determine if this is the case.

7. The authors note in the Discussion that the beating waveform changes dramatically in fluid with higher viscosity. Yet, if external dissipation plays such a small role how can this be rationalized?

---

## [Author Response]

Essential revisions:1. It is not completely clear from the manuscript what the configuration of the sperm is with respect to the glass slide where the head is tethered. What is the orientation of the cells with respect to the slide, and in which plane are the deformations measured? (from above or from the side?) We would expect that different configurations may lead to slightly different waveforms. In particular, we are surprised that the mean shapes shown in figure 2(a) have a net asymmetry which is observed in nearly all the cells: could this have to do with the relative configuration of the flagellum with respect to the surface?

Rodent sperm heads are flat and have falciform (hook-like) shapes. In our experiments with mouse sperm, the heads are tethered on one of their flat sides to the glass slide at the bottom of the imaging chamber. Imaging is done from above the sperm cell. The tethering renders the sperm head nearly immobile. As explained further below, the plane of the flagellar beat is approximately planar and parallel to the glass slide surface. Our calculations are restricted to the head, mid-piece and principal piece as these can be clearly resolved in the images. The end-piece (the distal 15% of the cell body) is neglected.

Intrinsic net asymmetry in flagellar beating is well known in sperm in many mammalian species, even when uncapacitated. Our observation that the mean shape is curved with a anti-hook (ventral) concave shape is consistent with the observations of Woolley (2003) that, in mouse sperm, the flagellum bends at the neck more on the ventral side than on the other.

Revisions

a. A new paragraph has been inserted in the “Tethering and imaging” subsection in the Materials and methods that summarizes the comments above on how the orientation of the cell relative to the microscope.

b. A figure is also included in a supplementary Appendix that shows a schematic of the side-view of the sperm beating plane parallel to the glass slide.

c. The comments above on the asymmetry of the mean shape have been inserted at the end of the second paragraph in the subsection, “POD enables identification of beat cycles”, in Results.

2. The experiments are done with flagella very near a no-slip surface, since the cells are chemically adhered to the chamber boundary. Yet, the authors use resistive force theory for filaments in free space, without any reference to the nearby no-slip surface. As the rate of energy dissipation near the surface will be considerably larger than estimated by RFT, it is possible that some (or much, or perhaps all) of the additional dissipation found by the authors is actually within the fluid and simply not accounted for by RFT. Thus, all of the calculations must be redone with the appropriate Blake tensor for stokeslets near a no-slip wall before the results can be considered definitive. The paper must also more carefully illustrate and quantify the proximity of the flagella to the surface in order to make these calculations precise. Absent this analysis, the claims of the paper do not stand up to scrutiny.

We acknowledge that the proximity to the wall is an important issue and address this comment in detail below. All the results presented in the revised manuscript are now with the distance from the wall taken into account. We find that this revised calculation does not alter the primary conclusions in our paper that internal dissipation – either due to passive internal friction or due to a mechanism associated with the motors – could be significant in sperm.

In our detailed response below, we discuss (a) the evidence from literature on which our estimate of the distance from the wall is based, (b) the RFT calculation with the wall distance taken into account and the resulting changes in the friction coefficients, and (c) how these changes leave our primary conclusions unaffected.

a. Wall distance. If the bottom of the sperm body were to be in direct contact with the wall everywhere, hydrodynamic frictional resistance will be very large. In that extreme case, flagellar beating would be expected to be significantly attenuated compared to the beating of freely swimming observed away from the wall. However, in all our samples where cells are tethered at their heads, the sperm continue to

beat their flagella freely.

A detailed study by Woolley (2003) showed that freely swimming mouse sperm exhibit “preferential capture” by walls. They are hydrodynamically drawn to walls and then stabilize with the left sides of their flat heads held against the surface. If a sperm arrives at a wall with its right-side next to the surface, it quickly moves away from the wall after a few flagellar beats and will be eventually captured in its stable orientation. Sperm following this “left-side rule” exhibit planar beating in a plane that is parallel to the wall. This beating is clearly resolvable within the focal width of the microscope. On the other hand, those that approach the wall with the right sides of their heads parallel to the wall exhibit considerable motion out of the focal plane. Scanning electron microscopy further revealed that the plane of the left-side of the flat head makes an angle less than 180°◦ with the flagellum at the neck. This appears to enable sperm following the left-side rule to stabilize planar beating parallel to the wall.

In our samples, the heads are chemically tethered to the wall. While we did observe a few cells adhere on their right sides, these cells exhibited non-planar beating out of the focal plane. Author response image 1 (a) and (b) sketch the expected geometry of the mouse sperm cell with its intrinsic head-tail angle at the neck, when the left side of its head is tethered to the wall. In this orientation, the flagellum beats in a plane parallel to the wall and its centreline is at a distance equal to the neck radius, *h* = *a*_n_, from the wall, where *a*_n_ is the radius of the neck. The head-neck angle is the same as the angle made by the head axis with the wall, θ ≈ aN/ℓH, where ℓH is the length of the head. From literature measurements of mouse sperm dimensions, *a*_N_ ≈ 0.6 *µ*m and ℓH≈ 5 μm, which gives *θ* ≈ 0.1 rad or 6°.

**Author response image 1. respfig1:** (**a**) Schematic showing the mouse sperm body as viewed from its dorsal side when the left side of its head is against the wall. The intrinsic angle made by the head with the flagellum at the neck enables planar beating when the left side of the head is against the wall. The red and green lines indicate the axes of the head and the tail. In this orientation, the flagellum beats in a plane parallel to the wall and its centreline is at a distance equal to the neck radius ,*a*n, from the wall. If there had been no angle at the neck, the tip of the tail would be at a height of *h*_T_ = *h*_max_. (**b**) The angle (in radians) of the head axis to the wall, *θ* ≈ *a*n*/* ℓh.

We note here that, in the process of recalculating the wall effect, we discovered that we had previously mistakenly used the width of 2.7 *µ*m of the basal region of the head as the neck radius and a radius at the tail end of 0.65 *µ*m. The correct value of the neck radius, *a*_n_, is 0.57 *µ*m, and the radius of the filament at the tail end, *a*_t_, is the axonemal radius, 0.18 *µ*m.

*b. Resistive Force Theory with wall distance* Katz et al. (1975) obtained the following Resistive Force Theory (RFT) approximations for the tangential and normal friction coefficients for planar motion of a slender body parallel to a wall, at a distance of *h* from the wall are(1)ζtwall=2πμln2h/a;ζnwall=4πμln2h/a

These coefficients have previously been used in a number of studies, notably by J^´^’ulicher and co-workers (Riedel-Kruse et al., 2007a) for analyzing experimental data on wall-tethered sperm and, more recently, by Mondal et al. (2020), for tethered axonemes isolated from cilia. We have now used these coefficients in all calculations along with *h* = *a*_n_ = 0.6 *µ*m, and the linear radial taper,(2)a(s)=(aN−aT)L−sl−sN+aTwhere *a*_n_ is the radius at *s* = *s*_n_ at the neck and *a*_t_ = 0.65 *µ*m is the radius of the axoneme at *s* = *L* at the tip of the tail.

In the original manuscript, we had used the following RFT coefficients for motion in bulk fluid well away from a wall (Lighthill, 1976):(3)ζtbulk=2πμln2q/a;ζnbulk=4πμln2q/a+1/2where *q* = 0.09*λ*, *λ* is the wavelength which can be approximated the flagellar length, *L* = 120 *µ*m. As noted above, we had also calculated the friction coefficients with *a*_n_ = 2.7 *µ*m. Author response image 2 compares the ratios of the friction coefficients near the wall and in the bulk(4)Wt=ζtwallζtbulk=ln(0.18L/a′)ln(2H/a);Wt=ζnwallζnbulk=ln (0.18L/a′)=1/2ln(2H/a)where *a*^0^ is the radius profile used in the original submission. Further, the ratio of the normal coefficient to the tangential coefficient, *γ* = *ζ_n_/ζ_t_
*is 2 for all *s* in the case of the Wall-RFT coefficients. With the bulk RFT coefficients in the original manuscript, this ratio is close to 1.8 and only varies weakly with *s*. Hence, the change in *γ* with the Wall-RFT coefficients is relatively smaller than the change in *ζ_t_*.

**Author response image 2. respfig2:** Ratio of the Wall-RFT coefficients in revised manuscript to the bulk coefficients in the original submission.

*c. Changes to results* The changes in the RFT coefficients directly impact the calculation of the hydrodynamic dissipation density,

phd=ζt(vt2+ζvn2), (5)

from the experimentally measured velocity components, *v_t_
*and *v_n_*. The change in *p^h^
*is mostly due to the change in *ζ_t_*, since the change in *γ* is relatively small. In the calculation of the total instantaneous hydrodynamic power dissipation, phd=∫ststphdds,and its cycle-average, the changes in *ζ_t_
*are weighted more strongly by the squares of the larger velocities towards the tail end. The revised values for the total hydrodynamic dissipation are about 2.5 times as large as before for all the samples (Author response image 3).

The change in *p*^hd^ affects the calculation of the active power density, *p*^a^, directly:(6)pa=∈˙−ps−phd−pid

Here, the densities of elastic storage rate, ˙, and the internal dissipation rate, *p*^id^, are unaffected by the changes in the RFT coefficients. The rate of mechanical work done by the passive material stress across the cross section,

ps=∂(v•F)∂s+∂(w•M)∂a. (7)

In this equation, only F depends on the RFT coefficients, since(8)F(s,t)=∫sLfh(s,t)ds′=∫SLζt(vt+γvn)ds′

**Author response image 3. respfig3:** Comparison of revised values of the cycle-means of (i) hydrodynamic dissipation, (ii) motor dissipation, (iii) active power input, and (iv) the proportion of hydrodynamic dissipation out of the total dissipation against their values in the original manuscript for WT (green symbols) and KO (yellow symbols) samples, with ηn = 0 (top panel) and 103 Pa s µm^4^(bottom panel): the values in the original submission are along the horizontal axes and the revised values are along the vertical axes.

A scaling analysis can be used to understand the relative contributions of the different terms in the energy balance that we use to determine *p*^a^. The elastic terms scale as *κ*_n_ ℓ^2^*w k*^4^, where *κ*_n_ is the elastic stiffness at the neck, ℓ is the beat amplitude and *w* and *k* are, respectively, the angular frequency and wavenumber corresponding to the travelling bending wave in the flagellum. If internal dissipation due to bending friction is included, those terms scale as *η*_n_ ℓ^2^*w*^2^*k*^4^, where *η*_n_ is the internal friction coefficient at the neck. The hydrodynamic contribution scales as *ζ_t,_*_n_ ℓ^2^*w*^2^. The relative magnitudes of the elastic and internal frictional contributions over the hydrodynamic contributions in the energy balance thus scale as *κ*_n_*k*^4^*/*(*ζ_t,_*_n_*w*) and *η*_n_*k*^4^*/ζ_t,_*_n_, respectively.

With *ζ_t,_*_n_ = 4*π µ/*ln(4*h*_n_*/a*_n_) when Wall-RFT is used, and with *κ*_n_ = 7 × 10^4^ Pa *µ*m^4^, , *η*_n_ = 10^3^ Pa s *µ*m^4^, *µ* = 10^−3^ Pa s, a beat frequency of *f* = 7 Hz, and a wavelength of about *L* = 100 *µ*m, the elastic and internal friction contributions in the energy balance are, respectively, 2.7 and 1.7 times larger than the contribution from hydrodynamic power. Since these terms dominating the energy balance remain unaffected by any changes in the RFT coefficients, *p*^a^ is less sensitive to changes in the RFT coefficients than the hydrodynamic dissipation itself. The motor dissipation and motor power input are determined from *p*^a^, and these are also not strongly influenced by the changes (Author response image 3).

As a result, while the numerical values of our results in Figures 3, 4–B and C, and 5 of the manuscript have changed, the key original conclusions still stand: that, in either the WT or *Crisp2* KO mouse lines we have studied, (i) there is significant motor dissipation within the axoneme, and (ii) the total internal dissipation within the flagellum is significantly larger than the external hydrodynamic dissipation.

Revisions

a. The Wall-RFT coefficients have been included in the Theoretical Modeling section in the paper, along with literature references.

b. The correct values of the neck radius, *a*_n_, is 0.57 *µ*m , and the radius of the filament at the tail end, *a*_t_, is the axonemal radius, 0.18 *µ*mhave been specified in the paragraph before the Results section.

c. All data in Figures3, 4 and 5 in the revised manuscript are with Wall RFT. The discussion on lines 359–382 associated with Figure 5 (C) of the relative magnitudes of the different dissipations has taken into account the increased hydrodynamic dissipation.

d. The second paragraph in the “Tethering and imaging” subsection in the Materials and methods section summarizes the discussion of tethering of the left-side of the heads in (a) in the response above.

e. A supplementary Appendix now includes the discussion in (a) in the response above as a separate section titled “Planar beating of tethered moue sperm”. A figure similar to Author response image 1 is also included. This figure shows the head shape when viewed with the left-side attached to the wall, the neck diameter, the schematic of the cell viewed side-on, showing the distance of the beating plane from the wall and the angle at the neck that enables beating parallel to the wall when the left-side of the head is tethered to the wall.

A related point is the need to understand the effect of tethering the cell on its kinematics and energetics? In other words, do the conclusions still hold for freely swimming cells?

We firstly note that our observations with tethered sperm are relevant for in their own right. Mammalian sperm, in vivo, are known to swim close to the oviductal wall. Uncapacitated sperm (such as the samples in our work) are known to also bind tightly at their heads to the oviductal epithelia while continuing to beat with their flagella oriented away from the wall (Ardon et al., 2016). While freely swimming in bulk fluids is more common in aquatic species that fertilize by broadcast spawning, in mammalian species, sperm localized near a wall is closer to physiological conditions. From the applied perspective understanding the behaviour of sperm near walls is also of considerable interest for developing artificial reproduction technologies that use microfluidic devices for sorting and separating sperm.

The reviewers have already pointed out the increased strength and anisotropy of the hydrodynamic resistance near a wall. The tethering constraint results in an additional force and torque being imposed at the head. We can certainly expect that these different external loading conditions will cause the kinematics of tethered sperm to be different from those of free swimmers , either at a wall or in bulk fluid. Nevertheless, we expect that our conclusions that most of the dynein power input is actually spent to overcome internal dissipation within the motors and the rest of the passive material will still be valid for free swimmers. We discuss the reasons for this in detail below in our responses to the reviewers’ queries (in Comments 7 and 8) on how our work relates to current theoretical understanding of sperm waveforms and the effect of increasing medium viscosity on the waveforms.

Revisions

The effect of tethering is discussed in the Discussion section in the revised manuscript along with the effects of fluid viscosity and wall proximity.

3. Is there any evidence of 3D dynamics? Some recent experiments with human sperm have suggested that sperm beats can take place in 3D (Gadelha et al., Science Advances 2020). As the model in the paper is 2D, this could also affect the energy balance.

We firstly note that there have been other studies that have used imaging of tethered sperm and cilia to understand axonemal dynamics (Mondal et al., 2020; RiedelKruse et al., 2007b). We have also taken advantage of the nearly planar beating that many mammalian sperm exhibit near walls. In our experiments, out-of-plane excursions in the resolved portion of the tail appear limited to less that 2 *µ*m. Nearly planar beating of the mid-piece and principal piece is evidenced by the fact that these portions of the flagellum remained in focus at all times in our samples. The

total depth of field(9)dtot=λnNA2+neMNAis 1.2 *µ*m in our microscope, where *λ* = 0.7 *µ*m is the mean wavelength of the incident light, *n* = 1 is the refractive index of the air medium between the coverslip and the objective lens, *M* = 20 is the lateral magnification, NA = 0.7 is the numerical aperture of the objective, and *e* = 0.65 *µ*m is the resolution of the detector in the image plane. After taking into account uncertainties due to the spread in incident wavelengths and other factors, we estimate that the depth of field cannot be larger than 2 *µ*m. We have verified this independently in calibration experiments using 5 *µ*m diameter spherical particles adhered to the bottom surface.

A maximum vertical deviation of around 2 *µ*m is about 10% of the mean amplitude of the in-plane beating of around 20 *µ*m. Therefore, neglecting such deviations can be expected to contribute errors of around 10% in the velocity components, curvature and rate of curvature. These errors propagate quadratically when calculating energetic quantities. The resulting error due to non-planarity in beating cannot therefore be larger than 1%, which is considerably smaller than the natural fluctuations in the beating patterns within a single sample and the sample-to-sample variations.

Revisions

a. The subsection on Tethering and Imaging in Materials and methods now includes as statement that out of plane deviations are less than 2 *µ*m.

b. The discussion above of the near planarity of the beating is now included in a

supplementary Appendix.

4. The authors should examine the work of K.E. Machin ["The control and synchronization of flagellar movement"], Proc. Roy. Soc. B 158, 88 (1963), which provided the first theoretical formalism to study active moment generation within beating flagella based on examining the difference between known force contributions from viscous dissipation and elastic bending. It seems that this same kind of analysis could be done here to identify directly the non-viscous contribution, rather than having to postulate a particular form.Stated another way: Why not try to estimate the active power density directly from the active moment density, which could be calculated from the moment balance of equation (4) where all the other terms are known? This would provide a direct estimate of the active power. The force balance could then be used to estimate the internal friction, which would then no longer rely on an assumed value for the internal friction coefficient. In fact, this could be used to obtain an estimate for that coefficient.

We missed citing Machin’s seminal work on flagellar waveforms and have corrected this oversight in the revised manuscript. However, estimating *both* the unknown internal friction coefficient and the unknown active power distribution (or the active moment density, *m*^a^) from the same set of data is problematic.

In the Kirchhoff model, the force F in the linear momentum equation (*i.e* the force balance) is the Lagrangian multiplier enforcing the inextensibility and nonshearability constraints. Since the net active force at any cross section is zero and there are no non-hydrodynamic external forces acting on the flagellum, the force balance for the tail region is:(10)fh+∂F∂s=0

The hydrodynamic force density , f^h^, is calculated from the observed kinematics using RFT. We therefore integrate this equation to calculate F at any cross section

in the filament as:(11)F (s,t)=∫slfh(s,,t)ds′

In other words, F is completely known from f^h^, and it cannot provide any further information on internal friction.

It is, in principle, possible to use Eqn. 4 for the conservation of angular momentum to calculate the active moment density, *m*^a^. The hydrodynamic moment, *m*^h^, is negligible relative to other terms and there are no non-hydrodynamic external moments, *m*^e^, that act on the tail. Splitting the passive internal moment M into elastic and dissipative contributions, the angular momentum equation for the flagellum reduces

to:(12)ma+t × F+ ∂Mel∂s+ ∂Mid∂s=0

In this equation, the tangent vector t is known from experiments. We also have the curvature and curvature rates to be used in the constitutive equations for M^el^ and M^id^, and as well as an estimate for the elastic stiffness, *κ*_n_, for mouse sperm from literature. We can use F determined from the previous equation.

However, *both* the active moment, *m*^a^, as well as the internal friction coefficient, *η*_n_, are unknown. It is not possible to use the same equation to determine these *independently of each other*. We have the same situation even while using the energy balance. We have, therefore, performed a parameter-sweep in *η*_n_, with values ranging from 0 to well above the scaling estimate of 10^3^. In the Discussion, we have pointed out the need for performing independent experiments for microrheological characterization of the flagellar material.

Revisions

In the Introduction, the start of the fourth paragraph discussing extracting forces and energetics from the measured beat patterns explicitly states: “Our approach for calculating forces and energetics from the measured beating patterns stems from ideas discussed originally by Machin….”

5. The paper addresses in detail the use of Chebyshev fitting methods for the filaments, but does not appear to address the physical boundary conditions one would expect on elastic objects (particularly at the free end), involving the vanishing of moments and forces. Unlike, for example, the biharmonic eigenfunctions of simple elastic filament dynamics which are tailored to those boundary conditions [see, e.g. Goldstein, Powers, Wiggins, PRL 80, 5232 (1998)], it is not clear how the Chebyshev functions satisfy those conditions. Some explanation is needed.

We thank the reviewers for bringing these interesting eigenfunctions of the hyperdiffusive operator for elastica to our attention and will explore using them in future.

Here, however, we have a composite body with free ends. The head region is treated as a rigid body, rotating about the tether point. The tail region is flexible and viscoelastic. Chebyshev polynomials are fitted only to through the tail region. Internal stresses are expected to be non-zero and continuous across the neck junction.

We do not apply the *dynamical* force- and torque-free conditions at the head and tail ends when fitting a smooth curve through the raw pixel data for the centerlines. Instead, we only place geometric and kinematic constraints. We ensure that, in the head region, the fitted tangent angle function satisfies rigid-body rotational kinematics. For the Chebyshev fits through the tail at any time instant, we ensure *C*^2^-continuity of the tangent angle profile across the neck. We have used the method of undetermined coefficients to impose these continuity constraints on the Chebyshev polynomial coefficients. The procedure is described in the subsection on Data Processing in Materials and methods.

The tail region is further complicated by the fact that the distal end of the tail often fades in and out of optical resolution because it is moving fast and is also very thin. We only have reliable data for what we refer to as the imaged-tail region, *s* ∈ [*s*_n_*,s*_t_], where *s*_t_ ≈ 0.85*L*. We therefore do not enforce any boundary conditions at the *imaged*-tail end (*s*_t_ = 0.85*L*). We find that enforcing a zero elastic moment (*i.e.* zero curvature) at *s*_t_ 6 = *L* produces unsatisfactory curve-fits to the experimental centerlines. Such curves tend to be too flat at the tail end.

We do, however, account for the force- and torque-free conditions at the free head and tail ends when developing the equations for extracting the internal dynamics and energetics from the measured kinematics. Firstly, the passive internal force in the tail is formally calculated from the hydrodynamic force distribution as(13)f (s,t)=∫sLfh(s′,t)ds′

This satisfies the condition that F(*L,t*) = 0. However, since we do not have motion data for *s > s*_t_, we neglect the contribution to the hydrodynamic forces from *s > s*_t_. This effectively amounts to applying the force-free condition at *s* = *s*_t_.

Secondly, the force- and torque-free conditions at the head end are applied when calculating the instantaneous power dissipated against the hydrodynamic and tethering forces on the head. The point-wise energy balance for the *rigid* head region (which cannot elastically deform to store energy) is:(14)0=phd+pe+pswhere *p*^hd^ and *p*^e^ are the hydrodynamic dissipation and tethering power densities and *p*^s^ = *∂/∂s*(v · F + *ω M*). Integrating this over the head-region at any time, and applying the boundary conditions at the free end, gives:(15)pHhd= pHe=(vo ∙ F0+ ω0M0)− (vN ∙ Fn+ ωNMN)= −(vN ∙ FN+ ωNMN)

Since stresses are continuous across the neck junction, we equate F_N_ and *M*_N_ in the equation above to those calculated with Eqn. (13) for F earlier and with the viscoelastic constitutive equation for *M* in the tail region. The *C*^2^-continuity of the fitted tangent angle profile at the neck ensures that the velocities, v_N_ and *ω*_N_, are uniquely defined at *s*_n_.

Thirdly, the force- and torque-free conditions at the tail end are applied to calculate the instantaneous power balance across the entire tail region. For the (untethered and viscoelastic) tail, the point-wise energy balance is:

∈˙=pa+phd+pid+ps. (16)

Integrating over the tail region, we obtain,(17)E˙=pa+phd+pid+(vL•FL+ωLML)−(vN•FN+ωNMN)

The free-end boundary condition is used to set F*_L_* = 0 and *M_L_
*= 0. Using Eqn. (15) earlier, the neck contribution can be replaced in terms of the total power dissipation by the head, *P*_h_^d^ = *P*_h_^hd^ + *P*_h_^e^. Further, we split the integral of *p*^a^ into contributions from its negative and positive parts by defining the motor dissipation and motor input as follows:(18)pmd(t)=∫sNLmin(pa,0)ds;pmi(t)=∫sNLmax(pa,0)ds. 

Substituting these in Eqn. (17) and rearranging, we obtain (noting that dissipations are negative in our sign convention)(19)pmi=E˙−phd−pid−pmd−pHd

In other words, the motor input provided is either used for elastic storage or to overcome all the contributions to dissipation.

This shows that the energy balance equations we have used in our calculations are formally consistent with the free-end conditions. It also shows that we have effectively neglected contributions from the non-imaged tail end. We do not, however, expect significant qualitative changes due to this approximation.

Revisions

a. Key details of our approach to determining the energetics from the measured kinematics have been brought forward from the Materials and methods section into the Theoretical Modeling section and described more clearly. The description also discusses how the boundary conditions are incorporated into the equations for the energetics.

b. The section on conservation equations in a supplementary Appendix has been augmented with the equations for the head and tail region.

6. If indeed internal dissipation dominates, that would suggest that essentially all prior theoretical approaches to calculating sperm waveforms must be quantitatively in error by very large factors. It would be very appropriate for the authors to examine some of those theoretical works to determine if this is the case.

We have now considerably revised the Discussion section to consider observations in the light of the current theoretical understanding of axonemal dynamics and sperm waveforms.

Several ideas have been presented in the past for the generation of the beating patterns by the axoneme. In a landmark study, Riedel-Kruse et al. (2007b) compared the predictions of many of these with experimental observations of planar beating in bull sperm that were either head-tethered or swimming freely in circles for long adjacent to a glass-slide wall. It was shown that the best agreement with experiments is obtained with the sliding-control model of Ju¨licher and co-workers (Camalet et al., 1999; Camalet and Ju¨licher, 2000). In this model (in the notation of the current paper), the active moment is related to the local internal shear and shear rate through an equation of the form *m*^a^ = *K γ* + *λ∂γ/∂t* where *γ* is the local shear strain. In the parlance of control theory, this model proposes that motors are regulated by the location deformation through a mechanism that follows a proportionalderivative control logic. More recently, Mondal et al. (2020) suggested a variant with proportional-integral control logic instead *i.e* where *m*^a^ + *β ∂m*^a^*/∂t* = *K γ*.

In either case, when the equation for regulation of the active moment is coupled with the equations for the rest of the passive material of the flagellum, an oscillatory instability emerges in certain ranges of the controller constants. This triggers a travelling wave that propagates down the filament and beating patterns similar to experimental data are obtained. It is further found with these models that the controller constants to achieve oscillations are negative, indicating that the active moment exerted by the dynein motors is down-regulated by the load exerted back on the motors due to the local shear deformation in the filament and its time rate of change. This also appears to be consistent with the recent experimental finding that dynein motors are always primed to deliver forces on microtubules but are inhibited when a curvature wave passes through their location (Lin and Nicastro, 2018).

It is possible that regulation of *m*^a^ could more generally described by an equation of the form, *m*^a^ + *β ∂m*^a^*/∂t* = *K γ* + *λ∂γ/∂t*, which corresponds to proportional-integral-derivative (PID) control. Such regulation of *m*^a^ immediately means that, when stable travelling waves are generated, the local rotation rate, *ω* (which is proportional to *∂γ/∂t*) will be systematically out of phase with *m*^a^, as is indeed observed in Figure 6. There will necessarily, therefore, be phases in each cycle when the two variables will be of opposite sign and *p*^a^ = *m*^a^*ω* will always be negative in those phases.

The mechanical work done back on the motors during such phases by the passive elements of the filament must be quickly dissipated in some form, since the motors cannot store the energy that is received nor reconvert it back to ATP. What, then, is the internal mechanism behind this additional dissipation? Riedel-Kruse et al. pointed out that the sliding-control model had to allow for relative sliding between microtubules at the basal end to obtain experimental agreement and that frictional resistance to basal shearing is important for the model to predict stable oscillations. Mondal et al. analyzed axonemes isolated by demembranating *Chlamydomonas* cilia and found that external hydrodynamic friction is too small to explain the stable beating pattern observed. They then showed that their sliding-control model predicts stable oscillations when coupled with equations that include passive filament elasticity and internal frictional resistance to the *shear* deformation rate. These sources of internal friction are not modeled in the present study, where we have treated the flagellum as an *unshearable* Kirchhoff rod. As Figure 5 (A) in the manuscript shows, we find that, if internal friction is absent or insufficient, then the observed motion would mean that, for a significant duration of the mean cycle, the filament may as a whole be driving the motors backward. While this unphysical picture is eliminated when a sufficiently high internal friction coefficient is used, we still observe motor dissipation due to *m*^a^ and *ω* being out of phase with one another.

The key point is that, while some or all of these different frictional contributions may be necessary for an internally-driven filament to oscillate stably, if the local regulation of the active moment in general follows PID logic, then the out-of-phase moment and local deformation rate will lead to phases of negative active power, irrespective of the nature of internal or external friction. This points to the existence of a separate dissipative mechanism associated with the dynein motors themselves.

There is already evidence that dyneins can dissipate energy locally. It is known that dynein motors can cycle through conformational changes driven by ATP binding and hydrolysis even when not driving microtubule sliding (Kon et al., 2005). Opticaltweezer experiments on dyneins bound to static microtubules have further shown that dyneins can steadily be driven in the reverse along the microtubule by an external load by forces larger than the stall force for these motors (Gennerich et al., 2007). The force required is more than that required to move unbound motors at the same velocity. This work done to drive the motors backward must be dissipated locally by a mechanism other than just the hydrodynamic frictional resistance of the motors to motion. Our results show that such motor dissipation can be a large part of the energy budget within the flagellum.

Revisions

The revised Discussion includes the response above, along with Figure 6.

7. The authors note in the Discussion that the beating waveform changes dramatically in fluid with higher viscosity. Yet, if external dissipation plays such a small role how can this be rationalized?

As noted in our response to the previous comment, even close to a wall, viscous dissipation appears to play a minor role in our experiments as well as those conducted with demembranated cilia (Mondal et al., 2020), *in aqueous media*. As pointed out in the original manuscript, *if the kinematics remain unchanged*, an increase in viscosity increases in a proportional increase in the hydrodynamic dissipation. However, as discussed above, the flagellar waveform emerges from the tight coupling of the active-viscoelastic axonemal dynamics to the passive viscoelasticity of the rest filament. Therefore, any change to the external mechanical environment – whether it be due to viscosity, or the presence of bounding surfaces, or the exertion of nonhydrodynamic tethering forces or moments – can lead to non-trivial changes in the emergent waveform.

For instance, Mondal et al. have shown that, when external hydrodynamic friction is negligible and internal resistance to shear deformation is taken into account, the governing equations for the passive material are diffusive in nature. When the external viscosity is large enough that hydrodynamic effects are not negligible, the equations governing the dynamics have a hyperdiffusive character. Therefore, one can expect that, the system may transition from one kind of qualitative behaviour to another with an increase in viscosity.

Revisions

The revised Discussion includes most of the comments above. We have not included the last paragraph above on the change in the nature of equations in the model of Mondal et al.